



# Fe(II) stability in seawater.

Mark J. Hopwood[1], Carolina Santana-González[2], Julian Gallego-Urrea[3], Nicolas Sanchez[4], Eric P. Achterberg[1], Murat V. Ardelan[4], Martha Gledhill[1], Melchor González-Dávila[2], Linn Hoffmann[5], Øystein Leiknes[4], Juana Magdalena Santana-Casiano[2], Tatiana M. Tsagaraki[6] and David Turner[3]

*Correspondence to:* Mark J. Hopwood (mhopwood@geomar.de)

1 GEOMAR Helmholtz Centre for Ocean Research Kiel, Germany
2 Instituto de Oceanografía y Cambio Global, IOCAG, Universidad de Las Palmas de Gran Canaria, ULPGC, Las Palmas, Spain
3 University of Gothenburg, Sweden
4 Norwegian University of Science and Technology, Trondheim, Norway
5 University of Otago, Dunedin, New Zealand
6 Department of Biological Sciences, University of Bergen, Norway

## Abstract

The speciation of dissolved iron (DFe) in the ocean is widely assumed to consist exclusively of Fe(III)-ligand complexes. Yet in most aqueous environments a poorly defined fraction of DFe also exists as Fe(II). Here we deploy flow injection analysis to measure in-situ Fe(II) concentrations during a series of mesocosm/microcosm experiments in coastal environments in addition to the decay rate of this Fe(II) when moved into the dark. During 5 mesocosm/microcosm experiments in Svalbard and Patagonia, where dissolved (0.2 µm) Fe and Fe(II) were quantified simultaneously, Fe(II) constituted 24-65% of DFe suggesting that Fe(II) was a large fraction of the DFe pool. When this Fe(II) was allowed to decay in the dark, the vast majority of measured oxidation rate constants were retarded relative to calculated constants derived from ambient temperature, salinity, pH and dissolved $O_2$. The oxidation rates of Fe(II) spikes added to Atlantic seawater more closely matched calculated rate constants. The difference between observed and theoretical decay rates in Svalbard and Patagonia was most pronounced at Fe(II) concentrations <2 nM and attributed to a stabilising effect of cellular exudates upon Fe(II). This enhanced stability of Fe(II) under post-bloom conditions, and the existence of such a high fraction of DFe as Fe(II), challenges the assumption that DFe speciation is dominated by ligand bound-Fe(III) species.

## 1. Introduction

The micronutrient iron (Fe) limits marine primary production across much of the surface ocean (Kolber et al., 1994; Martin et al., 1990; Martin and Fitzwater, 1988). Fe is required for the synthesis of the photosynthetic apparatus of autotrophs (Geider and Laroche, 1994), is an essential element in the enzyme nitrogenase required for $N_2$ fixation (Moore et al., 2009), and is important for phosphorous (P) acquisition from dissolved organic P compounds as part of the enzyme alkaline



phosphatase (Mahaffey et al., 2014). Fe is thus one of the key environmental control factors, or 'drivers', that concurrently regulate marine microbial community structure and productivity (Boyd et al., 2010; Tagliabue et al., 2017). The distribution of dissolved Fe (DFe) in the ocean (Tagliabue et al., 2017; Schlitzer et al., 2018) and the magnitude of the dominant atmospheric (Mahowald et al., 2005; Conway and John, 2014), hydrothermal (Tagliabue et al., 2010; Resing et al., 2015) and

shelf sources (Elrod et al., 2004; Severmann et al., 2010) are now moderately well constrained. Furthermore, dissolved Fe(III) speciation has also been explored in depth and it is evident that Fe(III)-binding ligands are a major control on the concentration and distribution of DFe in the ocean (Van Den Berg, 1995; Gledhill and Buck, 2012; Hunter and Boyd, 2007). Ligands (L), small organic molecules capable of complexing Fe(III), can maintain DFe concentrations of up to ~1-2 nM in oxic seawater which is an order of magnitude greater than the inorganic solubility of Fe(III) under saline, oxic conditions

(Liu and Millero, 1999, 2002). Characterising these ligands in terms of their concentrations and affinity for Fe(III) was therefore a major objective over the past two decades using a variety of related titration techniques (Gledhill and Van Den Berg, 1994; Rue and Bruland, 1995; Hawkes et al., 2013). 99% of DFe in the ocean is hypothesized to be present as Fe(III)-L complexes (Gledhill and Buck, 2012) and this observation explicitly or implicitly underpins the formulation of DFe in global marine biogeochemical models (Tagliabue et al., 2016).

There are however two specific environments in which this widely quoted "99%" statistic is incorrect. The first is oxygen minimum zones, where low $O_2$ concentrations extend the half-life of Fe(II) with respect to oxidation and thus permit high nanomolar concentrations of Fe(II) to accumulate in the water column accounting for up to 100% of DFe (Landing and Bruland, 1987; Lohan and Bruland, 2008; Chever et al., 2015). The second is surface waters where photochemical processes

initiate the redox cycling of DFe and permit measurable (>0.2 nM) concentrations of dissolved Fe(II) to exist in spite of rapid oxidation rates. Fe(II) is reported to account for 20% of surface DFe concentrations in the Baltic (Breitbarth et al., 2009), 12-14% in the Pacific (Hansard et al., 2009), and 5-65% in the South Atlantic and Southern Ocean (Bowie et al., 2002; Sarthou et al., 2011). A significant fraction of DFe is therefore likely present globally as Fe(II) in oxic surface waters. Yet oceanographic sampling of surface waters using rosettes is a poorly suited method for the analysis of Fe(II)

concentrations where the half-life of Fe(II) is significantly less than the inevitable time delay between sample collection and analysis (Hansard et al., 2009).

There is thus a paucity of Fe(II) data in the literature due to the formidable logistical challenges in collecting and analysing this transient species at sea (Hansard et al., 2009; Sarthou et al., 2011). The kinetic availability of dissolved Fe(II) relative to

dissolved Fe(III) (Sunda et al., 2001), the positive effect of redox cycling maintaining DFe in solution in bioavailable forms- irrespective of whether Fe(II) itself is bioavailable- (Croot et al., 2001; Emmenegger et al., 2001), and the potentially widespread presence of Fe(II) as a high fraction of DFe in surface waters (O'Sullivan et al., 1991; Hansard et al., 2009; Sarthou et al., 2011) imply that redox cycling is an important feature of marine Fe biogeochemistry. Yet, as evidenced by over-use of the "99%" statistic, the presence of a fraction of DFe as Fe(II) in surface waters –exactly where most primary



production occurs- is widely overlooked. Here, in order to characterize the behaviour of Fe(II) in surface waters we adapted flow injection apparatus to measure in situ Fe(II) concentrations both in a series of mesocosm experiments (Gran Canaria, Patagonia, Svalbard) and in adjacent ambient waters covering a diverse range of physical and chemical properties.

## 2.1 Mesocosm set up and sampling

The setup for the same series of incubation experiments from which we discuss results here (Table 1) is reported in detail in a companion paper (Hopwood et al., 2018b). However, for ease of access, a shorter version is reproduced here. Note that previously a series of experiments in the Mediterranean ('MesoMed') was also included. During the Mediterranean experiments the rapid oxidation rate of Fe(II) precluded the determination of Fe(II) concentrations. Fe(II) concentrations were universally <0.2 nM and thus no Fe(II) results from the MesoMed experiments are presented herein.

Briefly, all experiments used coastal seawater which was either pumped from small boats deployed offshore, or from the end of a floating jetty. Two of the outdoor mesocosm experiments (MesoPat and MesoArc) were conducted using the same basic design in different locations. For these mesocosms, 10 identical 1000-1500 L tanks (high density polyethylene, HDPE) were filled ~95% full with coastal seawater passed through nylon mesh to remove mesozooplankton. Fresh zooplankton
(copepods) were collected at ~30 m by horizontal tows with a mesh net, stored overnight in 100 L containers and non-viable copepods removed by siphoning prior to making zooplankton additions to the mesocosm tanks. After filling the mesocosms, the freshly collected zooplankton were added to 5 of the tanks to create contrasting high/low grazing conditions. Macronutrients ($NO_3/NH_4$, $PO_4$ and Si) were added daily. Across both the 5-high and 5-low grazing tank treatments, a DOC gradient was created by addition of glucose to provide carbon at 0, 0.5, 1, 2 and 3 times the Redfield Ratio (Redfield, 1934)
of carbon with respect to added $PO_4$. At regular 1-2 day intervals throughout each experiment, mesocosm water was sampled through silicon tubing immediately after mixing of the tanks using plastic paddles with the first 2 L discarded in order to flush the sample tubing.

A 3[rd] outdoor mesocosm experiment (Taliarte, Gran Canaria, March 2016) used 8 cylindrical polyurethane bags with a depth
of approximately 3 m, a starting volume of ~8000 L and no lid or screen on top (for further details see Filella et al., 2018 and Hopwood et al., 2018a). After filling with coastal seawater the bags were allowed to stand for 4 days. A pH gradient across the 8 tanks was then induced (on day 0), by the addition of varying volumes of filtered, $pCO_2$ saturated seawater (treatments outlined Fig. S1 IV) using a custom-made distribution device (Riebesell et al., 2013). A single macronutrient addition was made on day 18.




| Label | Location | Month / year | Experiment duration / days | Manipulated drivers | Scale / L | Site | Design (Fig. S1) | Fe data available |
|---|---|---|---|---|---|---|---|---|
| MesoPat (Ocean Certain) Mesocosm | Comau fjord, Patagonia | November 2014 | 11 | DOC, grazing | 1000 | In-situ | I | Diurnal time series, Fe(II) decay experiments, XRF time series |
| MesoPat (Ocean Certain) Multistressor | Comau fjord, Patagonia | November 2014 | 8 | DOC, grazing, pH | 20 | Temperature controlled room | II | Fe(II) decay experiments, XRF time series |
| MesoPat (Ocean Certain) Microcosm | Comau fjord, Patagonia | November 2014 | 11 | DOC, grazing | 20 | Temperature controlled room | III | Fe(II) decay experiments, XRF time series |
| MesoArc (Ocean Certain) Mesocosm | Kongsfjorden, Svalbard | July 2015 | 12 | DOC, grazing | 1250 | In-situ | I | Fe(II) decay experiments, Diurnal time series, XRF time series |
| MesoArc (Ocean Certain) Multistressor | Kongsfjorden, Svalbard | July 2015 | 8 | DOC, grazing, pH | 20 | Temperature controlled room | II | Fe(II) decay experiments |
| Gran Canaria (The Future Ocean) Mesocosm | Taliarte Harbour, Gran Canaria | March 2016 | 28 | pCO$_2$ | 8000 | In-situ | IV | Mesocosm Fe(II) time series |

**Table 1A Details of experiments where Fe data were collected. Data from 6 separate experiments are presented, including 3 outdoor mesocosm experiments and 3 indoor microcosm/multistressor experiments. 'DOC' dissolved organic carbon, 'XRF' X-ray fluorescence spectroscopy.**





| Experiment | PAT (Patagonia) | ARC (Svalbard, Arctic) | Gran Canaria |
|---|---|---|---|
| **Mesocosm** | | | |
| Containers | HDPE 1000 L | HDPE 1250 L | Polyurethane 8000 L |
| Zooplankton treatment | Addition of 30 copepods $L^{-1}$ | Addition of 5 copepods $L^{-1}$ | NA |
| Macronutrient addition | Nitrogen was added as $NO_3$ | Nitrogen was added as $NH_4$ | Nitrogen was added as $NO_3$ |
| Macronutrient addition timing | Daily | Daily | Day 18 only |
| Macronutrients added (per addition) | 1.0 µM $NO_3$, 1.0 µM Si, 0.07 µM $PO_4$ | 1.12 µM $NO_3$, 1.2 µM Si, 0.07 µM $PO_4$ (11.4 µM Si added on day 1) | 3.1 µM $NO_3$, 1.5 µM Si, 0.2 µM $PO_4$ |
| Screening of initial seawater | No screening | Screening by 200 µm | Screening by 3 mm |
| **Multistressor** | | | |
| Containers | HDPE collapsible 20 L | HDPE collapsible 20 L | |
| Zooplankton treatment | Addition of 30 copepods $L^{-1}$ | Addition of 5 copepods $L^{-1}$ | |
| Light regime | 15 h light / 9 h dark | 24 h light | |
| Macronutrient addition | Same as Mesocosm | Same as Mesocosm | |
| Macronutrient addition timing | Daily | Daily | |
| Macronutrients added (per addition) | 1.0 µM $NO_3$, 1.0 µM Si, 0.07 µM $PO_4$ | 1.12 µM $NH_4$, 1.2 µM Si, 0.07 µM $PO_4$ | |
| pH post adjustment | 7.54±0.09 | 7.76±0.03 | |
| pH pre-adjustment | 7.91±0.01 | 8.27±0.18 | |
| Screening of initial seawater | Screening by 200 µm | Screening by 200 µm | |
| Temperature / ℃ | 13-18 | 4.0-7.0 | |
| **Microcosm** | | | |
| Containers | HDPE collapsible 20 L | | |
| Grazing treatment | Addition of 30 copepods $L^{-1}$ | | |
| Light regime | 15 h light / 9 h dark | | |
| Macronutrient addition timing | Daily | | |
| Macronutrient addition | Nitrogen was added as $NO_3$ | | |
| Macronutrients added (per addition) | 1.0 µM $NO_3$, 1.0 µM Si, 0.07 µM $PO_4$ | | |
| Screening of initial seawater | Screening by 200 µm | | |
| Temperature / ℃ | 14-17 | | |

**Table 1B Experiment details for each experiment. For a visual representation of experiment designs, the reader is referred to Supplementary Material. 'HDPE' high density polyethylene.**



## 2.2 Microcosm and multistressor set up and sampling

A 10-treatment microcosm mirroring the MesoPat 10 tank mesocosm (treatment design as per Fig. S1 I, but with $6 \times 20$ L containers per treatment rather than a single HDPE tank) and two 16-treatment multistressor experiments (Fig. S1 II) were also conducted as part of the Ocean Certain project, using artificial lighting in temperature-controlled rooms (Table 1, Fig.

S1). Coastal seawater, filtered through nylon mesh, was used to fill 20 L HDPE collapsible containers. The 20 L containers were arranged on custom made racks with a light intensity of 80 µmol quanta m$^{-2}$ s$^{-1}$, approximating that at ~3 m depth. A diurnal light regime representing spring/summer light conditions at each fieldsite was used and the tanks were agitated daily and after any additions (e.g. glucose, acid or macronutrient solutions) in order to ensure a homogeneous distribution of dissolved components. In all 20 L scale experiments, macronutrients were added daily. One 20 L container from each

treatment set was 'harvested' for sample water each sampling day.

The experimental matrix used for the two Ocean Certain multistressor experiments duplicated the Ocean Certain mesocosm design, with an additional pH manipulation: ambient and low pH. The pH of 'low' pH treatments was adjusted by a single addition of HCl (trace metal grade) on day 0 only with pH measured prior to and after the addition (Table 1). Sample water

from 20 L collapsible containers was extracted using a plastic syringe and silicon tubing which was mounted through the lid of each collapsible container.

Throughout, where changes in mesocosms/microcosms are plotted against time, 'day 0' is defined as the day the experimental gradient (zooplankton, DOC, pH, pCO$_2$) was imposed. Time prior to day 0 was intentionally introduced during

some experiments to allow water to equilibrate with ambient physical conditions after mesocosm filling. Fe(II) concentration varies on diurnal timescales and thus during each experiment where a time series of Fe(II) or DFe concentration was measured, sample collection and analysis occurred at the same time each day.

## 2.3 Chemical analysis

### Trace elements

Trace metal low density polyethylene (LDPE, Nalgene) bottles were prepared via a three stage washing procedure: 1 day in detergent, 1 week in 1.2 M HCl, 1 week in 1.2 M HNO$_3$. TdFe samples were collected without filtration in trace metal clean 125 mL LDPE bottles. DFe samples were collected in 0.5 or 1 L trace metal clean LDPE bottles and then filtered through acid-rinsed 0.2 µm filters (PTFE, Millipore) using a peristaltic pump (Minipuls 3, Gilson) into trace metal clean 125 mL LDPE bottles within 4 h of sample collection. TdFe and DFe samples were then acidified to pH <2.0 by the addition of HCl

(150 µL, UpA grade, Romil) and stored for 6 months prior to analysis. Samples were then diluted using 1 M distilled HNO$_3$ (SpA grade, Romil, distilled using a sub-boiling PFA distillation system, DST-1000, Savillex) and subsequently analyzed by high resolution inductively coupled plasma-mass spectrometry (HR-ICP-MS, ELEMENT XR, ThermoFisherScientific) with



calibration by standard addition. To verify the accuracy of Fe measurements the Certified Reference Materials NASS-7 and CASS-6 were analysed after the same dilution procedure with the measured Fe concentration in close agreement with certified values (6.21 ± 0.77 nM certified 6.29 ± 0.47 nM, and 26.6 ± 0.71 nM certified 27.9 ± 2.1 nM). The analytical blank was 0.13 nM Fe. The field blank (de-ionized, MilliQ, water handled and filtered as if a sample in the field) was ~0.5 nM and

varied slightly between mesocosms, yet was always <16% of DFe concentration.

Fe(II) samples (unfiltered) were collected in trace metal clean 50 or 125 mL LDPE bottles, transferred to a clean laboratory and analyzed via flow injection analysis (FIA) using luminol chemiluminescence without preconcentration (Croot and Laan, 2002) exactly as per Hopwood et al., (2017a). Fe(II) samples during the Ocean Certain experiments were analysed

immediately after sub-sampling from each individual mesocosm/multistressor container. In Gran Canaria, prior to sampling, 10 µL 6 M HCl (Hiperpur-Plus) was added to the LDPE bottles in order to maintain the sampled seawater at pH 6 and thus minimize oxidation of Fe(II) between sample collection and analysis; a modification outlined by Hansard and Landing (2009). Fe(II) was then quantified within 2 h of sample collection. In all cases Fe(II) was calibrated by standard additions (normally from 0.1-2 nM) using 100 or 600 µM stock solutions. Stock solutions were prepared from ammonium Fe(II)

sulfate hexahydrate (Sigma-Aldrich), acidified with 0.01 M HCl and stored in the dark. A diluted Fe(II) stock solution (1-2 µM) was prepared daily. The detection limit varied slightly between FIA runs from 90 pM (Gran Canaria) to 200 pM (MesoArc/MesoPat).

Wavelength dispersive X-ray fluorescence (WDXRF) was conducted on triplicates of particulate samples collected by

filtering 500 mL of seawater through 0.6 µm polycarbonate filters. After air-drying overnight, samples were stored in PetriSlide boxes at room temperature until analysis at the University of Bergen (Norway). Analysis via WDXRF spectroscopy was exactly as described by (Paulino et al., 2013) using a S4 Pioneer (Bruker-AXS, Karlsruhe, Germany).

**Macronutrients and chlorophyll a**

Dissolved macronutrient concentrations (nitrate, phosphate, silicic acid; filtered at 0.45 µm) were measured spectrophotometrically the same day as sample collection (Hansen and Koroleff, 2007). For experiments in Crete, phosphate concentrations were determined using the 'magic' method (Rimmelin and Moutin, 2005) because of the ultralow concentrations. Nutrient detection limits inevitably varied slightly between the different mesocosm/microcosm/multistressor experiments, however this does not adversely affect the discussion of results herein. Chlorophyll a was measured by

fluorometry as per Welschmeyer (1994).

**Carbonate chemistry**

pH (except where stated otherwise, 'pH' refers to the total scale reported at 25ºC) was measured during the Gran Canaria mesocosm using the spectrophotometric technique of Clayton and Byrne (1993) with m-cresol purple in an automated



Sensorlab SP101-SM system and a 25ºC-thermostatted 1 cm flow-cell exactly as per González-Dávila et al., (2016). pH during the MesoPat experiments was measured similarly as per Gran Canaria using m-cresol. During MesoArc experiments pH was measured spectrophotometrically as per Reggiani et al., (2016). For calculation of Fe(II) oxidation rates constants as per Santana-Casiano et al., (2005), $pH_{free}$ was calculated from measured pH using the sulphate dissociation constants derived from Dickson (1990).

### 2.4 In-situ biogeochemical parameters

Fe(II) concentrations, and other key biogeochemical parameters, were measured in ambient surface (~10-20 cm depth) water at all three experiment locations; Comau fjord (Patagonia, November 2014), Kongsfjorden (Svalbard, June 2015) and Taliarte (Gran Canaria, March 2016). FIA apparatus was assembled in waterproof boxes on floating jetties. A 3 m PTFE sample line was then positioned to float approximately 1 m away from the jetty with seawater constantly pumped into the FIA using a peristaltic pump (MiniPuls 3, Gilson). The time delay between water inflow into the PTFE line and sample analysis was 60-120 s. The concentrations of complimentary chemical parameters (TdFe, DFe, DOC, pH) were determined on samples collected by hand using trace metal clean 1 L LDPE bottles. Salinity and temperature data was collected with a hand-held LF 325 conductivity meter (WTW) calibrated with KCl solution. To compare Fe(II)/$H_2O_2$ FIA data to discrete DFe/TdFe samples the mean of 7 FIA datapoints, corresponding to 14 minutes of sample intake and analysis time, was used.

### 2.5 Fe(II) decay experiments

A series of experiments was conducted in Patagonia, Svalbard, and under laboratory conditions in Kiel to investigate the change in Fe(II) concentration when water was moved from ambient light into the dark. In Patagonia and Svalbard, after collection of unfiltered 1-2 L samples in transparent 2 L HDPE containers, the PTFE FIA sample line was placed into the sample bottle and continuous analysis for Fe(II) and $H_2O_2$ begun. After a stable chemiluminescence response was obtained (typically 2-4 min after first loading the sample), the sample bottle was moved to a Al foil lined dark laminar flow hood and analysis continued for >1 h or until Fe(II) concentration fell below the detection limit (~0.2 nM). The time at which the sample was moved into the dark was designated t = 0. Subsamples for the determination of DFe and TdFe were retained from this time point. Theoretical decay rate constants for these experiments were calculated using the formulation presented in Santana-Casiano et al., (2005) with measured pH, temperature, dissolved $O_2$ and salinity (see s3.5). Dissolved oxygen was measured using an Oxyminisensor (World Precision Instruments). Salinity and temperature for each experiment were measured using a hand-held LF 325 conductivity meter (WTW). Measured decay rates were determined, assuming pseudo-first order kinetics, from linear regression of ln[Fe(II)] for t 0-15 minutes.

### 2.6 Quantifying the potential for Fe contamination during a mesocosm experiment

During the MesoArc mesocosm (Svalbard) a 'bookkeeping' exercise was conducted for the mesocosm and multistressor experiments by the sub-sampling of all solutions added to the incubated seawater. Aqueous additions consisted of: HCl



solution (used to apply the pH gradient), macronutrient solution, glucose solution and zooplankton. A short (1-2 h) 1 M HCl (trace metal grade) leach was applied to equipment placed within the mesocosm and also to the HDPE mesocosm containers prior to filling to provide a quantitative estimate of 'leachable' Fe. Atmospheric deposition of Fe into the tanks when open was estimated by deploying open bottles of de-ionized water within the vicinity of the mesocosms for fixed time intervals of

1 h in triplicate on 3 occasions and recording the approximate extent of time when the mesocosm lids were removed.

## Results

### 3.1 'Bookkeeping' Fe additions for a 1000 L mesocosm experiment (Svalbard)

Assembling and maintaining mesocosm scale experiments under trace-element clean conditions is a logistically challenging exercise (e.g. Guieu et al., 2010) and thus it was desirable to conduct a thorough assessment of the extent to which Fe

concentrations were subject to inadvertent increases during at least one experiment. It could not normally be determined directly and reliably how much inadvertent contamination occurred during the filling of the mesocosm containers because the filling procedure typically occurred over approximately 12-24 h duration. The Fe concentration in the near-shore water used to fill all of the mesocosms likely varied substantially over this time period due to wind and tidal water displacement in addition to variable surface runoff. Also, the mesocosms could not be sampled using trace metal clean conditions

immediately after (or during) filling.

In order to provide a rigorous assessment of Fe contamination during one experiment, Fe inputs were tracked in all additions to the MesoArc mesocosm and scaled to the mesocosm volume (initially 1200 L, declining by 15% over the experiment duration). Both DFe and TdFe were determined. However, DFe in seawater does not behave conservatively under most

circumstances due to the low solubility of Fe(III) and rapid scavenging of DFe from the water column (Landing and Bruland, 1987; Liu and Millero, 2002). TdFe concentration, on the other hand, can at least be used to assess the relative importance of 'inadvertent' Fe addition to the mesocosm. Volume weighting all additions (Table 2) to the MesoArc mesocosm experiment as per Eq. (1) using the mean (mid-experiment) mesocosm volume ($V_{mesocosm}$), and assuming that all additions were well mixed and TdFe behaved conservatively, produced a total mean concentration of 48 nM TdFe (Fig. 1). In addition to the

uncertain variability arising as the mesocosms were filled, approximately 8% (3.6 nM) of TdFe within the MesoArc mesocosms could be attributed to inadvertent addition (Fig. 1) over the experiment duration.

$$\text{Equation 1} \qquad \Delta[TdFe]_{mesocosm} = \frac{V_{addition}}{V_{addition} + V_{mesocosm}} \times [TdFe]_{addition}$$




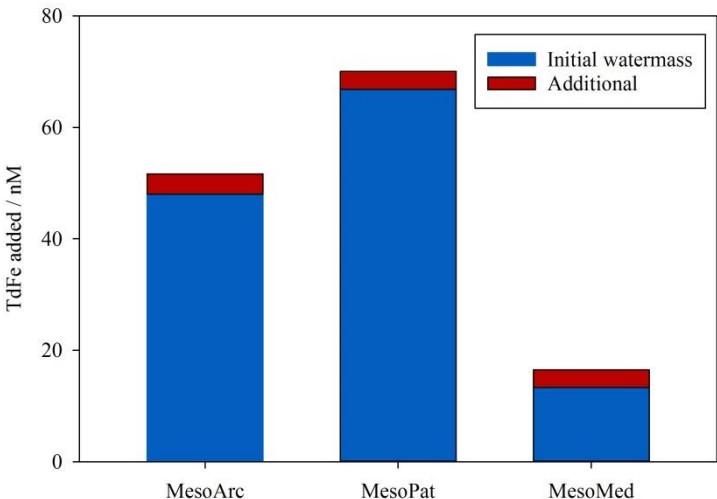

**Figure 1: Volume-weighted additions of TdFe to the same experimental design at three Ocean Certain mesocosm experiments. For MesoArc all inputs to the mesocosm were explicitly quantified. For MesoPat/MesoMed the initial water mass TdFe was quantified and TdFe inputs were adjusted as if the MesoArc experiment had been exactly duplicated with only the initial water mass**
**changed.**

When MesoArc is compared to the two other mesocosms with a similar design (MesoPat and MesoMed) the TdFe inputs and

the relative contribution of inadvertent TdFe addition were: 66.9 nM TdFe with 4.8% arising from inadvertent addition for

MesoPat and 13.3 nM with 24% TdFe arising from inadvertent addition for MesoMed (Fig. 1). Systematic contamination

was in all cases a minor, yet measurable, source of TdFe for these inshore mesocosms. Strictly, the inadvertent input of TdFe

varied between different treatments within each mesocosm experiment due to, for example, the variable volume of glucose

solution used to create a DOC gradient (Table 1). However, these differences caused small or negligible changes in TdFe

addition. It is not anticipated that this small TdFe addition will have had any adverse effect on the Fe redox chemistry results

presented herein for the Arctic and Patagonia experiments. As an additional precaution, sub-samples for Fe(II) analysis or

decay experiments were always collected when the mesocosms had been untouched (i.e. no sampling or additions) for >12 h,

thus Fe(II) species could not plausibly have been directly perturbed by any external manipulation of the

mesocosm/microcosm experiments.





| Fe source | TdFe addition / nM |
|---|---|
| Macronutrient spikes[a] | <0.01 |
| Glucose spikes[a] | <0.01 |
| Equipment added to mesocosms | 0.14 ± 0.04 |
| Zooplankton addition | 0.55 ± 0.01 |
| Atmospheric deposition | 0.87 ± 0.99 |
| Mesocosm plastic surfaces | 2.1 ± 0.54 |
| *Combined contamination and watermass variability during filling (percentage of initial TdFe)*[b] | *4-10% of initial [TdFe]* |

**Table 2. Total dissolvable Fe (TdFe) additions to the MesoArc mesocosm containers associated with sources other than the initial watermass.[a] These TdFe concentrations were measurable, but negligible when scaled to the mesocosm volume.[b] Based on TdFe measurements at time zero from the MesoPat multistressor/microcosm and DSi measurements on experiment day 0 or 1 from multiple mesocosms.**

5  **3.2 General trends in Fe biogeochemistry; the MesoArc (Svalbard) and MesoPat (Patagonia) mesocosms**

Before presenting the results of experiments designed to investigate the concentrations and stability of Fe(II), an over-view of Fe biogeochemistry within the different experiments is given. For all paired DFe/TdFe datapoints available during the MesoArc/MesoPat experiments (Fig. 2) the linear correlation between DFe and TdFe was not strong with most experiments maintaining a DFe concentration of 3-9 nM irrespective of TdFe. Curiously though, the correlation between DFe and TdFe

10  was stronger for MesoArc than for MesoPat (MesoPat $R^2$ 0.0022, gradient 0.0049 ± 0.014; MesoArc $R^2$ 0.48 gradient 0.036 ± 0.0073).



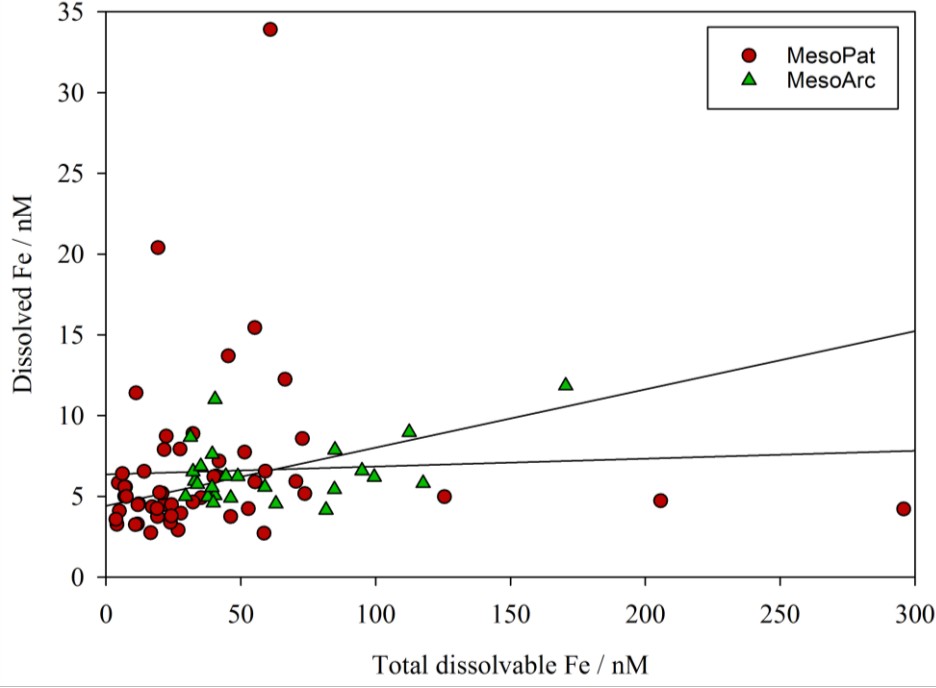

**Figure 2: DFe and TdFe from the MesoPat and MesoArc mesocosm and multistressor experiments where samples for both parameters were collected at the same timepoint. Linear regressions shown for MesoArc ($R^2$ 0.48 gradient 0.036 ± 0.0073) and MesoPat ($R^2$ 0.0022, gradient 0.0049 ± 0.014).**

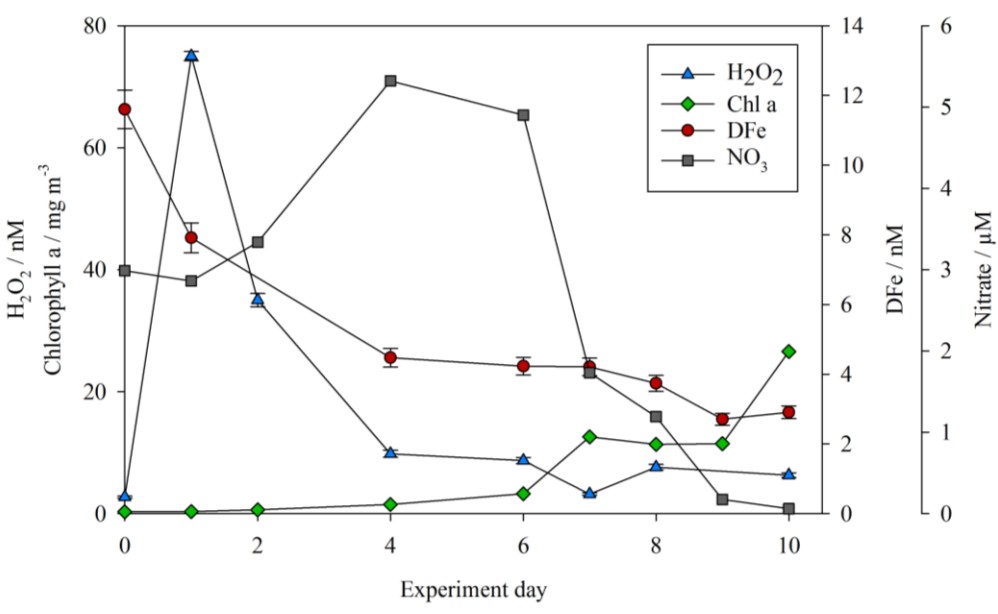

**Figure 3: DFe (red circles), hydrogen peroxide ($H_2O_2$, blue triangles), nitrate ($NO_3$, grey squares) and chlorophyll a (green diamonds) for the baseline treatment (no DOC addition, no added zooplankton) during the MesoPat mesocosm.**




Concentrations of both DFe and $H_2O_2$ (as per Hopwood, 2018) were measured at the highest resolution for the baseline treatment (no DOC addition, no zooplankton addition) during the MesoPat mesocosm. The initial concentration of DFe and $H_2O_2$ was estimated by using a Go-Flo bottle to sample at a depth of 10 m in the fjord (at which approximate depth the mesocosms were filled from). The apparent rise in $H_2O_2$ between day 0 and day 1 (Fig. 3) likely reflects the result of

increased formation of $H_2O_2$ after pumping of water from ~10 m depth into containers at the surface. $NO_3$ was added daily (Table 1b), hence concentrations increased prior to the onset of a phytoplankton bloom. The decline in DFe likely reflects biological uptake and/or scavenging onto particle (>0.2 µm) or mesocosm container surfaces.

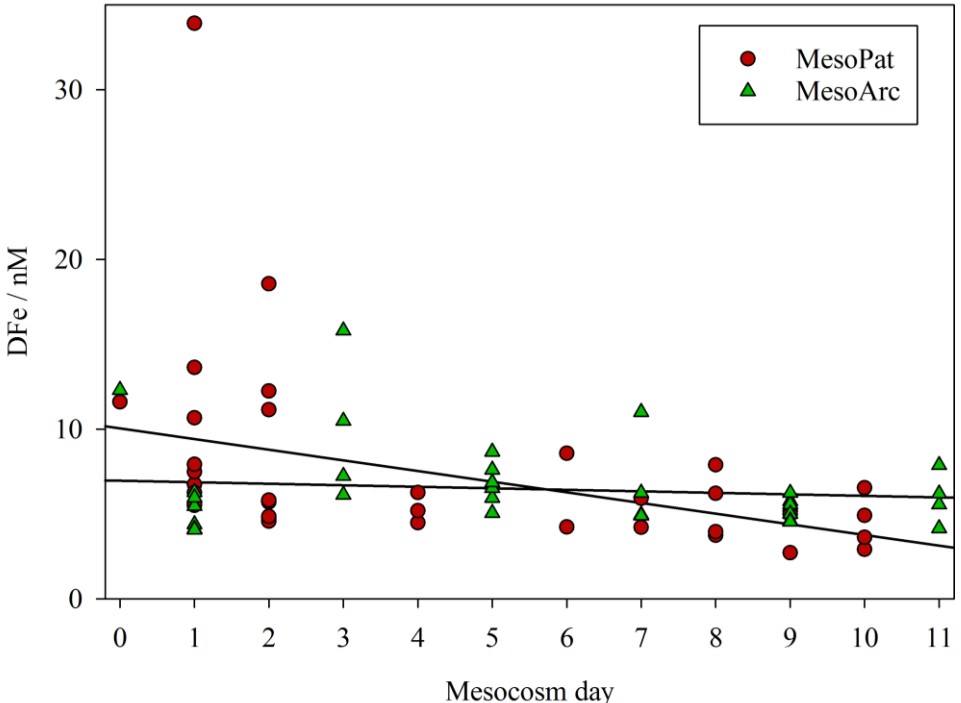

**Figure 4: DFe for all measurements made from the MesoPat (red circles) and MesoArc (green triangles) mesocosm experiments**
**against time. Linear regressions exclude the day 0 datapoints which were estimated from fjord water during mesocosm filling and**
**therefore were not strictly comparable to measurements within the mesocosms.**

Less frequent sampling for dissolved trace elements was available for treatments other than the 'baseline' no DOC/zooplankton addition treatment, but the decline in DFe during the MesoPat mesocosm was apparent across all measurements considered together (-0.63 ± 0.24 nM day$^{-1}$ derived from linear regression $R^2$ 0.16, Fig. 4). When all available

MesoArc DFe data was compiled similarly, the DFe concentration was steady over the duration of the mesocosm (-0.09 ± 0.13 nM day$^{-1}$ derived from linear regression $R^2$ 0.016, Fig. 4).

In addition to TdFe measurements from unfiltered water samples, particulate (>0.6 µm) Fe concentrations were also determined from wavelength dispersive X-ray fluorescence. WDXRF data were normalised to phosphorus (P) in order to



discuss trends in the elemental composition of particles and are thus presented as the Fe:P [mol Fe mol$^{-1}$ P] ratio. The initial Fe:P ratio in particles varied between the three mesocosm fieldsites: MesoMed 1.20 ± 0.34, MesoPat 0.34 ± 0.09 and MesoArc 0.62 ± 0.07. A similar trend however was observed during all experiments; a general decline in Fe:P across all treatments with time. Particulate Fe:P ratios on the final day of measurements was invariably lower than the initial ratio:

MesoMed 0.16 ± 0.04, MesoPat 0.09 ± 0.04, MicroPat 0.05 ± 0.01, MultiPat 0.07 ± 0.03, and MesoArc 0.17 ± 0.08. All of these ratios are high compared to literature values reported for offshore stations where the ratio ranged from 0.005 to 0.03 mol Fe mol$^{-1}$ P (Twining and Baines, 2013). However, this may simply reflect elasticity in Fe:P ratios which increase under high DFe conditions (Sunda et al., 1991; Sunda and Huntsman, 1995; Twining and Baines, 2013). Alternatively, it could reflect the inclusion of a large fraction of lithogenic material, which would be expected to have a higher Fe:P ratio than

biogenic material.

Particles from ambient waters outside the mesocosms were collected and analysed at the Patagonia and Svalbard fieldsites in order to assist in interpreting the temporal trend in Fe:P. Suspended particles from Kongsfjorden (Svalbard) exhibited a Fe:P ratio of 3.01 ± 0.06 mol Fe mol$^{-1}$ P and suspended particles in Comau fjord varied more widely with a mean ratio of 0.54 ±

0.41. Kongsfjorden surface waters are characterised by extremely high TdFe concentrations originating from particle rich meltwater plumes and thus the 3.0 Fe:P ratio can be considered to be a lithogenic signature. After ambient water was collected for the mesocosm experiments, the steady decline in particle Fe:P ratios throughout the experiments likely resulted partially from a settling or aggregation of lithogenic material after filling of the mesocosms. At the same time, a decline in the ratio of dissolved Fe:PO$_4$ during each experiment, due to the daily addition of PO$_4$ and minimal addition of new Fe, may

also have led to reduced Fe uptake relative to P.

### 3.4  Fe(II) time series (Gran Canaria)

A key focus of this work was to determine the fraction of DFe present as Fe(II). During the Gran Canaria mesocosm, a detailed time series of Fe(II) concentrations was conducted. The timing of sample collection was the same daily (14:30 UTC) in order to minimise the effect of changing light intensity over diurnal cycles on measured Fe(II) concentrations. Over

the duration of the Gran Canaria mesocosm, Fe(II) concentrations fell within the range 0.10-0.75 nM (Fig. 5 (a)). On the first measured day (day -2) Fe(II) ranged from 0.13 nM (mesocosm 7, 700 µatm pCO$_2$) to 0.63 nM (mesocosm 6, 1450 µatm pCO$_2$) with an overall mean concentration of 0.41 ± 0.12 nM. Generally, Fe(II) concentrations declined across all treatments from day 1 to 9. From day 9 to 20 strong variations were observed between treatments. Following nutrient addition on day 18, a phytoplankton bloom was evident in chlorophyll a data from day 19 or 20 with chlorophyll a peaking on day 21 or later

(Hopwood et al., 2018a). An increase in Fe(II) was then evident from days 20-29 under bloom and post-bloom conditions (Fig. 5 (a)).




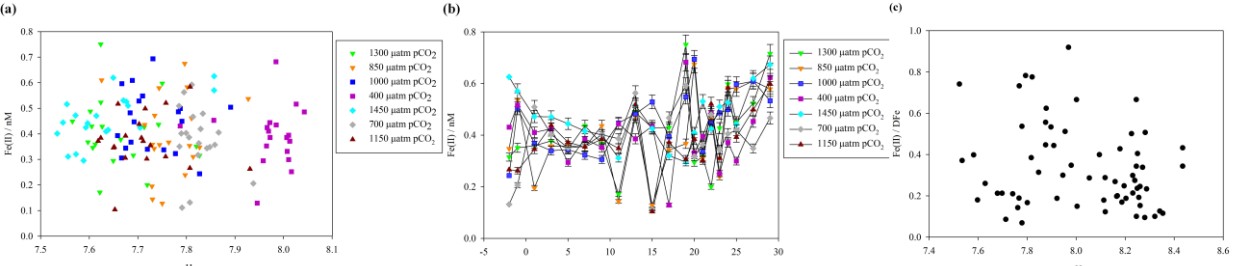

**Figure 5: (a) Fe(II) concentrations (unfiltered) during the Gran Canaria mesocosm plotted against measured mesocosm pH (b) Fe(II) concentrations over the duration of the Gran Canaria mesocosm experiment. The 550 µatm pCO$_2$ mesocosm was discontinued after leakage and exchange with surrounding seawater occurred on experiment day 3 and so no data is shown (c) Fe(II)/DFe for all available timepoints from MesoArc and MesoPat.**

Contrasting days 1 and 29, Fe(II) in all of the mesocosms except number 7 experienced a measurable increase in Fe(II) concentration (+0.4, +0.4, +0.2, +0.2, +0.2, 0.0 and +0.3 nM respectively from mesocosm number 2 to 8). Mesocosm 7 was also anomalous with respect to slow post-bloom nitrate drawdown and elevated H$_2$O$_2$ concentration (100 nM H$_2$O$_2$ greater than other treatments under post-bloom conditions (Hopwood et al., 2018a)). Overall, despite the large gradient in pCO$_2$ (400-1450 µatm and a corresponding measured pH range of 8.1-7.7), Fe(II) showed no significant correlation with pH (Pearson Product Moment Correlation p 0.32) (Fig. 5 (b)).

## 3.5 Fe(II) decay experiments (Patagonia and Svalbard)

During the Ocean Certain MesoArc and MesoPat experiments, a series of decay experiments (n = 79) was conducted to investigate the stability of in-situ Fe(II) concentrations. The 79 time points at the start of these experiments were made before water was moved from ambient lighting into the dark and can be considered as in-situ Fe(II) concentrations. Across the complete dataset, the properties known to affect the rate of Fe(II) oxidation in seawater varied over relatively large ranges for the various experiments; temperature 4.0-18°C, salinity 22.7-33.8, pH 7.46-8.44, 315-449 µM O$_2$, and 1-79 nM H$_2$O$_2$ (see Supplementary Material A). Initial Fe(II) concentrations ranged from 0.3-16 nM. Generally a decline in Fe(II) was observed immediately after transferring this sampled water to a dark box, yet this was not always the case. The Fe(II) concentration more often than not remained measurable (> 0.2 nM) for the entire duration of the decay experiment. One hour after the transfer of water from ambient conditions into the dark, Fe(II) was below detection on only 2 out of 79 occasions, and on average 55% of the initial Fe(II) concentration at t = 0 remained.

In order to account for the many physio-chemical parameters that affect Fe(II) oxidation rates, theoretical pseudo-first order rate constants (k') were calculated for each decay experiment (n = 79) using measured pH, salinity and temperature as per Eq. (2) (Santana-Casiano et al., 2005) where T is temperature (°K), pH is pH$_{free}$ and S is salinity (psu). O$_2$ saturation was calculated as per Garcia and Gordon (1992) and then k' was adjusted for measured O$_2$ concentrations as per Eq. (3). Measured rate constants (k$_{meas}$) were derived from the gradient of ln[Fe(II)] against time for each decay experiment from at



least 5 sequential datapoints (Fe(II) concentration was obtained at 2 minute intervals). One potential complication with calculating oxidation rates is that Fe(II) is oxidised via both $O_2$ and $H_2O_2$ in surface seawater (King et al., 1995; Millero and Sotolongo, 1989). Fortunately, the MesoPat and MesoArc experiments were notable for low $H_2O_2$ concentrations due to the enclosed HDPE containers used (Hopwood et al., 2018b) and therefore literature oxidation constants describing the

oxidation of Fe(II) via $O_2$ derived under low $H_2O_2$ conditions are particularly appropriate constants to use.

$$\text{Equation 2} \quad logk' = 35.407 - \left(6.7109 \times pH_{free}\right) + \left(0.5342 \times pH_{free}^2\right) - \left(\frac{5362.6}{T}\right)$$

$$-(0.04406 \times S^{0.5}) - (0.002847 \times S)$$

$$\text{Equation 3} \quad k = \frac{k'}{[O_2]}$$

The rate constant, k (Eq. 3), thus accounts for the major effect of variations between experiments of salinity, temperature, pH and $O_2$ in a single constant. Before comparing $k_{meas}$ and k, an estimate of the uncertainty should also be made as differences between the two values may arise due to the relatively large combined error from propagating the uncertainty in $S/T/pH_{free}/[O_2]$, and in analytical error on Fe(II) measurements. The accuracy of Fe(II) measurements is challenging to

quantify for a transient species with no appropriate reference material. In this case, the exact Fe(II) detection method used here was previously compared to another variation of the luminol chemiluminescence method (with pre-concentration, Hopwood et al., 2017) and $k_{meas}$ was determined with $\pm20\%$ difference between two methods. The uncertainty on $k_{meas}$ is therefore assumed to be $\pm20\%$ rather than the generally smaller uncertainty than can be calculated from linear regression of ln[Fe(II)]. The uncertainty in calculated k can be assessed by calculating the change resulting from the estimated uncertainty

on measured salinity ($\pm0.1$), temperature ($\pm0.5°C$), $pH_{free}$ ($\pm0.05$) and $O_2$ ($\pm10$ µM). The combined uncertainty is $\pm35\%$ for k. Reduced uncertainties are possible with closed thermostat systems where the uncertainty on all physical/chemical parameters ($S/T/pH/O_2$) would be significantly reduced, however our objective here was to measure the decay rates of in situ Fe(II) concentrations and thus the first priority was to commence measurements after sub-sampling rather than to stabilize physical/chemical conditions.

In order to further understand the cause of any systematic discrepancies in the dataset between measured $k_{meas}$ and calculated k, an additional set of experiments was conducted using aged South Atlantic seawater. This water was previously stored in 1 $m^3$ trace element clean HDPE containers for in excess of 1 year and was maintained in the dark at experimental temperature for 3 days prior to commencing any experiment. The background concentration of Fe(II) in this water was below detection

(<0.2 nM) and the initial DFe concentration relatively low ($0.98 \pm 0.39$ nM). In a series of 47 decay experiments, Fe(II) spikes of 2-8 nM were added and then the decay in the dark monitored as per the Arctic/Patagonia in-situ experiments.

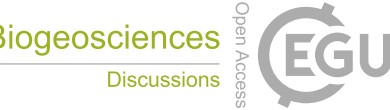



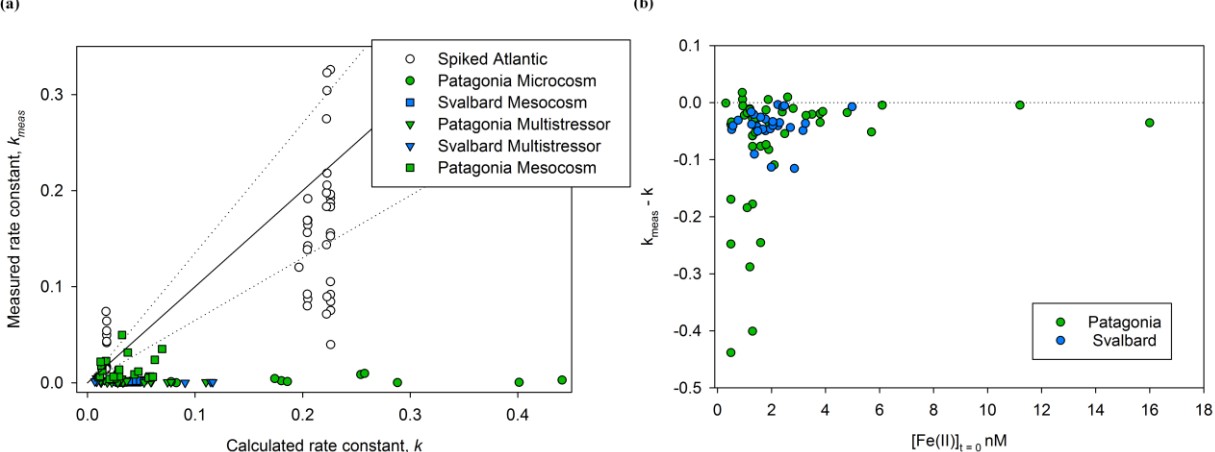

**Figure 6: A comparison of $k_{meas}$ and calculated k (both $M^{-1} min^{-1}$) for Fe(II) decay experiments. (a) Rate constants for Fe(II) decay experiments from Patagonia (green), Svalbard (blue) and spikes to aged Atlantic seawater (colourless) (b) The difference between observed and calculated values of k ($k_{meas}$-k) is shown against Fe(II) concentration at t = 0.**

Contrasting k with $k_{meas}$ (Fig. 6), it is immediately apparent that the Fe(II) present within Arctic/Patagonia experiments was generally much more stable than would be predicted for an equivalent inorganic spike of Fe(II) added to water with the same physical/chemical properties i.e. in most cases $k_{meas}$< k. Three plausible hypotheses can be conceived for this offset:

   i.   The measured rates here refer to relatively low initial Fe(II) concentrations (0.3-16 nM) compared to the concentrations at which rate constants have been derived (typically ~20-200 nM) and the difference arises simply because the rate constants are not calibrated for low nanomolar starting concentrations.

   ii.  There is 'dark' production of Fe(II) in the experiments i.e. on-going formation of Fe(II) counter-acts the first order decay of Fe(II) via oxidation.

   iii. The speciation of Fe(II) in seawater is more stable with respect to oxidation than the species for which the rate constants are calculated.

For the series of experiments using spikes of Fe(II) in South Atlantic seawater, $k_{meas}$ is consistently closer to k than for any in-situ experiments (Fig. 6a). Nevertheless, some datapoints for spiked South Atlantic seawater still fall outside the ±35% uncertainty boundary. As the spiked experiments closely matched the initial Fe(II) concentrations in the in-situ decay experiments, the higher Fe(II) concentrations generally used to establish the rate of Fe(II) decay in laboratory experiments cannot be the main explanation for a discrepancy between $k_{meas}$ and k, although it may be a minor contributing factor.

Calculating the difference between calculated and measured k (Δk), it is evident that the largest differences were associated with the lowest initial Fe(II) concentrations (Fig. 6b). This is consistent with both hypothesis II and III. Assuming that the dominant source of Fe(II) is photochemistry, the effects of both a secondary 'dark' Fe(II) source and a limited fraction of





Fe(II) existing in a more stable form with respect to oxidation would be most evident at the lowest initial Fe(II) concentration. Sources of Fe(II) other than photochemistry are plausible and may include, for example, zooplankton grazing due to the reduced pH and $O_2$ within organisms' guts (Nuester et al., 2014; Tang et al., 2011). Mesozooplankton addition was one of the three experimental variables manipulated during the Arctic/Patagonia experiments. However, no clear trend was evident with respect to the measured offset in k and the zooplankton addition status of the experiments. Mean $\Delta k \pm SD$ ($\times 10^{-2}$) for the high/low zooplankton treatments over all experiments were $4.66 \pm 5.79$ and $4.08 \pm 5.63$, respectively. A dependency of $\Delta k$ on the initial Fe(II) (Fig. 6b), with $[Fe(II)]_{t=0}$ likely very sensitive to multiple experimental factors such as the time of day that the sample was collected and the exact time delay between sample collection and the first timepoint for each Fe(II) decay experiment, would however make determining the relative importance of any other underlying causes challenging. In order to gain further insight into the potential role of zooplankton in Fe(II) release under dark conditions, a series of incubations was conducted with addition of the copepod *Calanus finmarchichus* to cultures of the diatom *Skeletonema costatum* (Hopwood et al., 2018b). No change in extracellular Fe(II) or $H_2O_2$ concentrations were evident across a gradient of copepods from 0-10 $L^{-1}$. Whilst this suggests the role of high/low zooplankton treatments was minimal in short-term changes to ambient Fe(II) concentrations, the potential release of Fe(II) by zooplankton may of course be species specific; different results may have been obtained with different zooplankton-prey combinations.

## 4 Discussion

### 4.1 Assessing the extent of Fe contamination within mesocosms

Whilst both DFe and TdFe inputs into any incubation experiment can be determined, DFe does not behave conservatively, is actively taken up by microorganisms and scavenged onto particle surfaces. Thus the relationship between TdFe and DFe is not a simple linear function (Fig. 2). The equilibrium concentration of Fe within particulate and dissolved phases depends on factors such as Fe(III) ligand, or more generally DOC, concentrations (Wagener et al., 2008) and particle loading (Bonnet and Guieu, 2004; Rogan et al., 2016). All of the incubation experiments herein were conducted using coastal or near-shore waters. This is reflected in the low salinities of the MesoPat (27.5-28.0) and MesoArc (33.7-33.8) mesocosms. Both of these fieldsites were fjords with high freshwater input. Comau fjord (Patagonia, MesoPat) is situated in a region with high annual rainfall and receives discharge from rivers including the River Vodudahue. Kongsfjorden (Svalbard, MesoArc) receives freshwater discharge from numerous meltwater fed streams and marine terminating glaciers in addition to melting ice. Correspondingly high DFe and TdFe concentrations were thereby found in surface waters; universally >4 nM DFe. The Gran Canaria (initial S 37.0) mesocosm cannot be considered to have had a coastal low salinity signature from large freshwater outflows, but was still conducted using near-shore waters which would generally be expected to contain higher Fe concentrations than offshore waters due to benthic sources of Fe (see, for example, Croot and Hunter, 2000). Despite the inshore basis of the MesoArc mesocosm, Fe contamination was a small, but significant, fraction of the TdFe added to the starting water (8%, 3.6 nM).





## 4.2 Fe speciation within the mesocosms

Throughout all of the MesoArc/MesoPat experiments, Fe(II) consistently constituted a large fraction of DFe (Table 4). The presence of 24-65% of DFe in mesocosms as Fe(II) is not unexpected, as the photoreduction of Fe(III) species by sunlight is well characterized (Barbeau, 2006; Wells et al., 1991). Yet it also raises questions about how Fe speciation is modelled in

these waters. DFe in the ocean is almost universally assumed to be characterised as "99% complexed by organic species"(Gledhill and Buck, 2012) on the basis of extensive research using voltammetric titrations to determine the strength and concentration of Fe binding ligands (Van Den Berg, 1995; Rue and Bruland, 1995). Yet these approaches exclusively measure Fe(III)-L species (Gledhill and Buck, 2012).

| Dataset | f [Fe(II)]/[DFe] | f [DFe]/[TdFe] | n |
|---|---|---|---|
| MesoArc mesocosm | 0.30 ± 0.14 | 0.15 ± 0.06 | 20 |
| MesoArc multistressor | 0.30 ± 0.17 | 0.07 ± 0.01 | 8 |
| Svalbard, ambient (light) | 0.11 ± 0.05 | <0.01 | 5 |
| MesoPat microcosm | 0.24 ± 0.14 | 0.76 ± 0.34 | 10 |
| MesoPat mesocosm | 0.65 ± 0.52 | 0.20 ± 0.17 | 22 |
| MesoPat multistressor | 0.47 ± 0.44 | 0.35 ± 0.30 | 15 |
| Patagonia, ambient (light) | 0.06 ± 0.04 | 0.12 ± 0.01 | 5 |
| Patagonia, ambient (dark) | 0.02 ± 0.00 | 0.15 ± 0.11 | 3 |

**Table 4. Fraction of dissolved Fe concentration ([DFe]) present as Fe(II), and fraction of total dissolvable Fe concentration**
**([TdFe]) present as DFe. n, number of datapoints. *ND*, not determined. All values are mean ± standard deviation.**

Here we should note that the method utilized during these incubation and diurnal experiments, flow injection analysis with a PTFE line inserted directly into the experiment water, is relatively well suited for establishing the in-situ concentration of Fe(II) (O'Sullivan et al., 1991). Such an experimental set up ensures no unnecessary delay is introduced between the collection and analysis of a sample. When using an opaque sampler, such as a Go-Flo bottle typically deployed at sea for

collection of trace element samples (Cutter and Bruland, 2012), the collection process inevitably displaces near-surface water from its ambient light conditions for a time period that constitutes >1 half-life of Fe(II) in warm, oxic seawater. Measured near-surface Fe(II) concentrations on samples from a rosette system would therefore always be expected to under-estimate in-situ Fe(II) concentrations (O'Sullivan et al., 1991).

Fe(II) concentration was also quantified in ambient waters adjacent to the mesocosms and found to constitute a lower fraction of DFe (2-11%). Most of the decay experiments, from which initial Fe(II) concentrations are reported in Table 4, were conducted at the end of mesocosm/microcosm experiments and thus it is not possible to assess the development of Fe(II) stability throughout the bloom in the Patagonia or Svalbard experiments. Nevertheless, the high fraction of DFe





present as Fe(II) in these experiments (Table 4) relative to that observed in ambient waters is consistent with the increase in Fe(II) concentrations observed in Gran Canaria after the initiation of the phytoplankton bloom (day 19 onwards, Fig. 5 (b)). The Patagonia/Svalbard experiments had macronutrient additions daily, whereas the Gran Canaria experiment had macronutrient addition only on day 18. The conditions within the Arctic/Patagonia experiments during the time period which

decay experiments were conducted were therefore typical of those during, or shortly after, a phytoplankton bloom. Whilst chlorophyll a was not quantified for ambient waters, for which Fe(II) data are reported in Table 4, sampling in Svalbard (July 2015) and Patagonia (November 2014) occured during low productivity phases relative to the annual cycle in primary production at these fieldsites (Hop et al., 2002; Iriarte et al., 2013).

### 4.3 Fe(II) decay experiments

Fe(II) oxidation rates are relatively well constrained in seawater with varying temperature, salinity, pH, $H_2O_2$ and $O_2$ concentration from extensive series of experiments where the change in concentration of an Fe(II) spike was monitored with time and the rate constants for oxidation with $O_2$ and $H_2O_2$ then derived from first order kinetics (e.g. King et al., 1995; Millero et al., 1987b). Whilst dissolved $O_2$ is the dominant oxidizing agent for Fe(II), $H_2O_2$ is also of importance as an Fe(II) oxidizing agent in surface seawater (González-Davila et al., 2005; King and Farlow, 2000; Millero and Sotolongo, 1989).

The unusually low concentration of $H_2O_2$ within the Patagonia and Svalbard experiments due to the enclosed HDPE mesocosm design and/or synthetic lighting (Hopwood et al., 2018b) was therefore fortunate from a mechanistic perspective as it allows the simplification that $O_2$ was the only major oxidising agent. The much lower $H_2O_2$ concentrations (1-79 nM) present, compared to ambient surface waters, during the Patagonia and Svalbard experiments should mean that Fe(II) decay rates during these experiments more closely match the oxidation rate constants used to derive Eq. 2 (which were derived for

low-$H_2O_2$ conditions).

The decay experiments reported here still however differ in two critical respects from controlled oxidation rate experiments used to derive rate constants. First, the speciation of Fe(II) may differ. It is debatable to what extent Fe(II)-L species, analogous to Fe(III)-L species, exist in surface marine waters due to the absence of reliable techniques to probe Fe(II)-

organic speciation (Statham et al., 2012), but there is consistent evidence that organic material affects Fe(II) oxidation rates (see below). Second, these decay experiments measure the change in Fe(II) concentration between light and dark conditions and not specifically the oxidation rate. If photochemical Fe(II) production was the sole source, and oxidation of Fe(II) via $H_2O_2$ and $O_2$ were the only Fe(II) sinks, then the decay rate measured here would approximate the oxidation rate determined under controlled laboratory conditions. However, there are possible biological sources of Fe(II) (Nuester et al., 2014; Sato et

al., 2007), the possibility of biological uptake of Fe(II) (Shaked and Lis, 2012) and cross-reactivity with other reactive trace species (e.g. reactive oxygen species and Cu, Croot and Heller, 2012) to consider. All of these complexities make Fe(II) more challenging to model in natural waters compared to controlled conditions. This is especially the case at low Fe(II)




concentrations relevant to the surface ocean where Fe(II) concentrations range from below detection up to ~1 nM (Gledhill and Van Den Berg, 1995; Hansard et al., 2009; Sarthou et al., 2011).

The high magnitude of Δk in some cases at low initial Fe(II) concentrations (Fig. 6) is consistent with the theory that Fe(II)

binding ligands are responsible for the observed stability of Fe(II) in some natural waters (Roy and Wells, 2011; Statham et al., 2012). The Fe(II)-binding capacity of any ligands present in a specific sample would be expected to become saturated as Fe(II) concentrations increased. The effect of Fe(II) ligands on the oxidation rate of an added Fe(II) spike would therefore become less evident as Fe(II) concentration increased because the fraction of Fe(II) present as Fe(II)-L species would decline i.e. $k_{meas}$ would converge with k. This has an important methodological implication. The effect of cellular exudates, or

natural organic material extracts, on Fe(II) oxidation rate is more often than not tested by adding reasonably high nanomolar Fe(II) spikes to solution and then following the Fe(II) decay with time (see, for example, Lee et al., 2017). By raising the initial Fe(II) concentration, such an approach may however systematically under-estimate the effect of organic material on Fe(II) stability at in-situ Fe(II) concentrations.

The effect of organic material on Fe(II) is difficult to generalize as organic compounds can accelerate, retard or have no apparent effect on Fe(II) oxidation rates via oxygen (Santana-Casiano et al., 2000). However, there are now sufficient studies of Fe(II) behaviour to distinguish between the broad effects of allochthonous and autochthonous material. Extracts from the green algae *Dunaliella tertiolecta* (Gonzalez et al., 2014), cyanobacteria *Synechococcus* (Samperio-Ramos et al., 2018b) and *Microcystis aeruginosa* (Lee et al., 2017), coccolithophore *Emiliania huxleyi* (Samperio-Ramos et al., 2018a), and diatoms

*Chaetoceros radicans* (Lee et al., 2017) and *Phaeodactylum tricornutum* (Santana-Casiano et al., 2014) have all been found to retard Fe(II) oxidation rates. Furthermore, the effect of cellular exudates on the reaction constant appears to scale with increasing total organic carbon (Samperio-Ramos et al., 2018b). This is also consistent with the release of Fe(II)-binding agents resulting in the formation of Fe(II)-L species with slower oxidation rates than inorganic Fe(II) speciation under specified physical/chemical conditions. In contrast to the stabilization apparent in some cellular exudates, allochthonous

material generally, although not universally, has the opposite effect with an acceleration of Fe(II) oxidation rates reported both in coastal environments (Lee et al., 2017) and using terrestrially derived organic leachates (Rose and Waite, 2003). The generally positive effects of cellular exudates on Fe(II) stability with respect to oxidation determined in single-species studies is consistent with the stability of Fe(II) observed in almost all experiments here (Fig. 6) and this suggests that microbial cellular exudates are indeed a stabilizing influence on Fe(II) concentrations at a broad scale in surface marine

environments. Stabilization of Fe(II) by freshly produced exudates could explain the sustained increase in Fe(II) concentrations across all $pCO_2$ treatments under post-bloom conditions in Gran Canaria and the high fraction of DFe present as Fe(II) during all Arctic/Patagonia experiments.

## 4.4 Conclusions

The existence of a high fraction (24-65%) of DFe as Fe(II) during mesocosm experiments, and the apparent stability of low concentrations of Fe(II) in these productive waters suggests that the classic characterisation of '99% of dissolved Fe existing as Fe(III)-L complexes' (Gledhill and Buck, 2012) is inadequate to describe DFe speciation in marine surface waters. Fe(III)-ligand complexes may overwhelmingly dominate Fe speciation in the ocean as a whole, but in sunlit surface waters a dynamic redox cycle operates maintaining considerable concentrations of Fe(II) in solution. The stabilizing effects on Fe(II) with respect to oxidation reported here were strongest at low (<2 nM) Fe(II) concentrations suggesting that the Fe(II) stabilization mechanism is caused by a process akin to complexation where the magnitude of the effect is capped by a factor other than physical conditions.

Exudates stabilizing Fe(II) may be a poorly characterized component of the aptly named 'ferrous wheel' (Kirchman, 1996; Strzepek et al., 2005) and contribute to the efficient recycling of DFe within marine surface waters. Irrespective of whether Fe(II) is more or less bioavailable relative to Fe(III), the formation of Fe(II) is a mechanism for increasing DFe and thus increasing DFe availability to biota. Mechanisms such as the stabilization of Fe(II) by cellular exudates during and after phytoplankton blooms may therefore facilitate DFe uptake to a greater extent than would be possible in the absence of Fe-redox cycling. Both Fe(III) and Fe(II) speciation and concentration must therefore be defined in order to understand the role of Fe as a driver of marine primary production.

## 4    Author Contributions

All authors contributed to the design of the study and the interpretation of data. MH, CS, JG, NS, ØL and TT conducted analytical work. MH coordinated the writing of the manuscript with input from other authors.

## 5    Acknowledgements

The Ocean Certain and KOSMOS/PLOCAN teams assisting with all aspects of experiment logistics and organisation are thanked sincerely for their efforts. Labview software for operating the $H_2O_2$ FIA system was designed by P Croot, M Heller, C Neill and W King. Financial aid from the European Commission (OCEAN-CERTAIN, FP7- ENV-2013-6.1-1; no: 603773), DFG (Collaborative Research Centre 754 Climate-Biogeochemistry Interactions in the Tropical Ocean) and Ministerio de Economía y Competitividad of the Spanish Government (EACFe project, CTM2014-52342-P) is gratefully acknowledged.

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
