# Peer review of "Fe(II) stability in coastal seawater during experiments in Patagonia, Svalbard and Gran Canaria."

_Biogeosciences, 2018_

## Referee Comment (RC1) · L. Gerringa (Referee) · 11 Nov 2018

The authors measured FeII and FeII oxidation under varying conditions. They conclude that FeII is more important than often claimed, since Fe is assumed to be for 99% in the III form and complexed by DFeIII-binding organic ligands. The residence time of FeII is longer than predicted since the oxidation is slower than predicted, at least the natural occurring Fe is. When Fe is added and above 2 nM, the oxidation rate is faster and resembling more theoretical oxidation rates. It is interesting, not really new. This subject does deserve attention. Experiments are well executed, data seems good and solid.

Meso and micro-cosms were sampled at different locations. The explanation of these experiments is very hard to follow. I am not sure all information is necessary, at least in the main manuscript. A simple table like table 1B is in my view enough. From

all these experiments FeII was measured and some samples were taken to observe FeII oxidation rates in the dark. In the end the explanation of different results hardly needs the differences between the experiments, the more reason to move text on the "cosms" to the supplementary information. The manuscript needs considerable improvements before it can be published. The authors missed one important publication Rijkenberg et al., 2006. In this paper the influence of organic ligands of FeIII and FeII on photo-reduction and oxidation of Fe are studied. In the present text it is assumed to be important, I agree, and the Rijkenberg publication can certainly help here instead of referring to papers where speculation on this subject can be found. One of the reasons that the text is hard to follow is the use of different names for the same locations/experiments (there are 5 locations: the Arctic (Svalbard), the Mediterranean (probably Crete), Patagonia, Gran Canaria and Kiel (mentioned only once and not in methods, but important for the discussion)). Since the setup is so complicated it is even advisable to mention experiments in the same sequence (helping the reader) and not as in the example given changing the sequence in one sentence: "MesoArc than for MesoPat (MesoPat R2 0.0022, gradient 0.0049 $\pm$ 0.014; MesoArc R." etc

The introduction tells the reader that she/he can expect mesocosm experiments in three places. Then section 2.1 starts and experiments were done in 4 places en in the result section experiments in Kiel appear to have happened too. Extra names like the ocean Certain project come out of the blue. First meso is used later without any explanation on page 14 Meso Med, MicroPat MultiPat are used. These last set meso micro and multi are much better suited because they indeed indicate which kind of experiments are meant. Why not use them immediately and define them properly?

Section 3.5 is very interesting but difficult to read and understand. Half of the text should be in methods, also be more clear about the Kiel experiments (at least I suspect the spiked Atlantic tests are done in Kiel). And it is not clear whether the different treatments had influence on the results.

In the discussion I miss apart from the Rijkenberg paper, discussing the influence of the

different sampling and measurement treatments and the different experimental conditions. Is the difference in time between sampling and analysis discussed, Gran Canaria is different from the others. Can this have had an effect on the results? See also above section 3.5. What is e-microcosm (in Suppl table),this is not explained, still here the largest differences between kmeas and kcal exist.

Detailed comments Sections 2.1 and 2.2 are hard to follow

Sentences like "Note that previously a series of experiments in the Mediterranean ('MesoMed') was also included." do not help. If it was previously, why do we bother here. Line 7-9 page3 section 2.1 seem out of place, this has nothing to do with setup and sampling. Line 13: 10 identical 1000-1500L tanks, 5 tanks got zooplankton. According to table 1A they all received copopods but the addition was different per location. I did not find figure S1, below text and pictures, there is a caption but no schematic figure of the experimental design. Line 26: can bags stand? Are bags mesocosms? In the next line the word tank is used, is this still the same thing? Section 2.2 What is a 10-treatment? Section 2.3,line 26 after cleaning, what happened with the bottles? Were they stored empty or filled, if so with what. Page 7: were the FeII bottles dark plastic? Line 9: Ocean certain is? It would be so much easier when the normally used names are used here too, meso/micro-Med-Arc-Pat . Section 2.5: What happened in Kiel? Line 18 tells us what happened in Patagonia and Svalbard (? Pat and ARC, Comau fjord- Kongsfjorden?) In 3.5 79 experiments are mentioned, that info belongs here. How many per location. Were they kept under ambient temperature, where is the laminar flowhood 3.1 why use the name Svalbard here and not MesoArc. Page 10 line 11, no glucose is mentioned in Table 1 Page 11 line 9: curiously..Why.is this curious, and why give relations that are not relations, for figures 2 and 4. For figure 4 it is not clear which line belongs to which mesocosm experiment. No idea what the journal guide lines are, but especially as in figure 3the './ 'is confusing like there is a ratio instead of the unit. Page 13 line 1 as per? Page 14: Meso Med, MicroPat MultiPat: have mercy on your reader! Suddenly new abbreviations. However, useful abbreviations that

should be used throughout the whole manuscript Lines 9-10: Is that to be expected? Reference needed here (also at line 16) 9-16, a lot of different names for the same sites 29: chlorophyll a Italic Figures 5 are too small. Lables and legend are impossible to read and the sequence in the legend is not logic, one legend might be enough for 5a and 5b. Perhaps make 5 c a separate figure. Be careful with ratio's. The high values are they due to low DFe? Figure 5 c is not mentioned in the text. Line26: I do not know whether there is a general decline as claimed here, the 1450 microatm perhaps does decreases but the 1300, 700 and 1150 microatm do not, so no general decline here. This figure is not suited to make such statements. An increase between days 20-29 ok. Page 15: line 6: what is number 7? Lines 6-9: not clear what the authors tell here? Is this still about figure 5b? The sixth mentioned number does not show an increase, not the seventh. They have different CO2, haven't they? Page 16, Line 11: which variation, give reference, do not force the reader to search in another of your papers to find out. Lines12-14: it depends what you mean with in situ and what you want to do with the k-values. With such an uncertainty one can wonder whether waiting for stabilisation would have been wiser. Lines 20 onwards: is this what happened in Kiel? Most of this belongs to the method section. Also the equations belong to the method section in my view. Page 17:I can add a few hypotheses: the aged Atlantic water was probably filtered, in any case no phytoplankton or copepods were present. The added Fe for certain saturated the organic ligands and thus this DFe was in an inorganic form, a colloidal or amorphous Fe-oxide or hydroxide. This is where the equations 2 and 3 were made for: inorganic Fe! So certainly this is other chemistry. I advise to read Rijkenberg et al., 2006 Geochimica et Cosmochimica Acta 70 (2006) 2790–2805; Enhancement and inhibition of iron photoreduction by individual ligands in open ocean seawater.

Page 18:line one, why would low FeII be the most stable? Discussion line 18: This can be read that TFe behaves conservatively. Why would DFe-TFe be linear, that is a strange idea. That is assuming all particles have the same properties. Page 19: table 4 why is mesopat meso and multistressor so different, this is not discussed. Why is the sequence different, why Svalbard whereas it is Arctic. The different names

makes it more difficult to understand. Lines 23-24: why was this not mentioned in the method section? Page 20 line 8-9: thus what is the conclusion? 4.3: line 16: according to the methods section artificial light was used in micro and multistressor but not in Mesocosm, so why mention artificial light here? Lines 21-25: read Rijkenberg, they saw the influence of ligands on Fe redox, of ligands binding Fe III en of a ligand binding FeII. That should be added in the discussion here. Page 21decay rates in the e-microcosm are different from the calculated k compared to the others, apart from low FeII at t=0? (low Fe(II) occurs also in other experiments) what is e-microcosm, what is different? Could that be an extra reason. Use the work of Rijkenberg et al in the discussion on page 21, they did not assume FeII ligands, they used one in their redox rate experiments.

Excel file temp in k, make capital, add start or initial also to the column name for FeII. The precision does not warrant the decimals shown with 35% uncertainty. What is an e-microcosm, why are the rates so high here. Add measured to k. No Kiel experiments here?

---

## Referee Comment (RC2) · Rose (Referee) · 7 Dec 2018

GENERAL COMMENTS

This manuscript addresses a topic that is relevant to the scope of Biogeosciences. There is a clear rationale for the work, the experiments have been carefully designed and executed, and the data analysis and interpretation are reasonable. However, I felt that the scope of the paper needs to be more accurately represented and that some details relating to the experiments and data analysis were missing. Overall, I believe that this manuscript should be published after revision.

SPECIFIC COMMENTS

1. The title of the manuscript is too broad, to the extent that I find it misleading. The manuscript does not directly address the issue of Fe(II) stability in seawater – there

are no measurements of thermodynamic constants (which the word "stability" implies), nor measurements or calculations of complex speciation, the underlying mechanisms are inferred or hypothesised rather than explicitly measured or tested, and the measurements are limited primarily to coastal seawater. This is all perfectly valid, but the manuscript really addresses iron redox speciation in coastal mesocosm experiments, and I would prefer to see a title more along these lines.

2. The assertions about "over-use of the "99%" statistic" (i.e. that "99% of DFe in the oceans is hypothesized to be present as Fe(III)-complexes" are subjective and I find this aspect of the Introduction to be overstated. It is true that "this observation explicitly or implicitly underpins the formulation of DFe in global marine biogeochemical models", and that the influence of Fe redox speciation is often ignored. The authors also provide a nice summary of compelling evidence that Fe(II) is important in "two specific environments". However, it does not automatically follow that the assumption that 99% of DFe is present as Fe(III)-complexes is invalid everywhere in the oceans, or that the "99% statistic" is "over-used". To make this assertion objectively would require something like a meta-analysis of the literature to quantify the number of papers that make this claim, and the proportion of those that make this claim incorrectly. In my opinion, it would be better to just present the evidence and let the reader decide if they think this is an "over-used" statistic. I would suggest that the authors review the Introduction to remove or tone down subjective statements and ensure that any assertions are supported by an appropriate number of references.

3. In the Introduction there is a strong focus on why Fe(II) is important, but the background about what is known in relation to the abundance and behaviour of Fe(II) in the ocean seems incomplete. For example, the growing body of work (including by some of the co-authors of this manuscript) around the influence of organic exudates from marine phytoplankton on Fe(II) oxidation kinetics is not mentioned in the Introduction, but this would seem critical to understanding much of the manuscript and its rationale. In addition, while there is a brief overview of Fe(II) dynamics in the photic zone and in

suboxic zones, it would also be useful to briefly review reports of Fe(II) measurements in other parts of the ocean.

4. Analysis of Fe(II) data was based on an assumption of pseudo-first order kinetics, but there are no details on whether this assumption was tested or verified.

5. I think it is highly problematic to exclude discussion of the Mesomed Fe(II) results from the manuscript because "Fe(II) concentrations were universally < 0.2 nM" (p. 3, lines 8-9). Given that you are arguing that Fe(II) is widespread and overlooked, excluding presentation of results from one set of mesocosms because Fe(II) was not measurable in those conditions could be perceived as cherry picking data. Again, I think this would be less of an issue if the scope of the manuscript as suggested by the title and Introduction was revised. If this is recast to make it clear that this is a study of Fe(II) dynamics in a discrete set of mesocosm experiments, then I think it is fine to mention the Mesomed experiments in this way without a detailed presentation of results. However, I think it is also important not to overlook these results in the discussion when generalising about Fe(II) behaviour.

6. The discussion about processes contributing to Fe(II) formation lacks mention of superoxide-mediated Fe reduction or other biological ferrireductase processes. This would seem remiss given that recent publications have suggested extracellular superoxide production may well be ubiquitous (e.g. Diaz et al., 2013, Widespread production of extracellular superoxide by heterotrophic bacteria, Science 340: 1223-1226) and is likely to influence Fe speciation (e.g. Rose, 2012, The influence of extracellular superoxide on iron redox chemistry and bioavailability to aquatic microorganisms, Frontiers in Microbiology 3:124).

7. The organisation of different aspects of the manuscript needs to be reviewed to ensure material is presented in the correct location. For example, the first paragraph of section 3.1 is discussion, not results. The second paragraph of section 3.1 is methods, not results. Details about measurement of hydrogen peroxide concentrations are not
provided in the methods section at all, but rather addressed only by the statement "as per Hopwood, 2018" in the results section.

8. P. 1, line 14. I suggest changing "exclusively" to "almost exclusively" or "primarily". It is not strictly correct to say that dissolved Fe speciation is assumed to consist exclusively of Fe(III)-L, as Fe' is generally also considered (although minor).

9. P. 2, lines 31-32. The argument that "the potentially widespread presence of Fe(II)" implies that "redox cycling is an important feature of marine Fe biogeochemistry" is a circular argument. The three cited papers do not show that Fe(II) is potentially widespread – they show that Fe(II) is persistent in certain specific environments and locations studied in this papers. I don't mean to be overly critical about this – I think Fe(II) is important and possibly overlooked – but I think it's important to be objective and precise.

10. P. 17, lines 9-11 and 19-21. This hypothesis is not plausible, in my opinion. A difference in rate constants between different Fe(II) concentrations could be related to a difference in chemical mechanism, but should be completely independent of calibration. Also, there are several studies of Fe(II) oxidation kinetics in seawater that have been conducted at low nanomolar concentrations such that there is a coherent mechanistic understanding (and ability to predict) Fe(II) rate constants from the low nanomolar range right through to the micromolar range.

TECHNICAL CORRECTIONS

11. P. 1, line 20. I suggest changing "retarded relative to" to "less than". Rates can be fast or slow, but rate constants are large or small.

12. P. 1, line 25. Please add a qualification to this sentence explaining under what conditions your work challenges these assumptions (e.g. in coastal surface waters?).

13. P. 2, line 8. Ligands are not necessarily small or organic. Perhaps could change this to "Organic ligands (L) capable of complexing Fe(III) can...".

14. P. 2, lines 24-26. This sentence seems like it belongs in the next paragraph... I can't see how this relates to the presence of Fe(II) in suboxic or photic zones.

15. P. 2, line 28. "There is a paucity of Fe(II) data..." – what sort of Fe(II) data?

16. P. 2, line 29. What do you mean by "kinetic availability"? Do you mean kinetic lability?

17. P. 2, lines 34-35. "as evidenced by over-use of the 99% statistic" – what evidence? No citations are provided and this is not tested robustly, as stated in point 2 above.

18. P. 3, lines 6-10. Following on from point 5 above, I find it confusing that some Mesomed results are included in the results, but no details are provided in the methods about these experiments, other than these couple of sentences. I think you need to treat this dataset in a similar way to the other mesososm results, namely describe the method details in full, and fully account for the Mesomed results in your discussion.

19. Tables 1A and 1B. It would make more sense to me to label these Table 1 and Table 2, as they show quite separate information. Furthermore, it would be useful to provide coordinates for the mesocosm locations in Table 1.

20. P. 6, line 4. Can you provide any information about the spectral quality of the lighting?

21. P. 6, line 25. Should this be "trace metal clean low density polyethylene" rather than "trace metal low density polyethylene"?

22. P. 7, line 22. Change "as described by (Paulino et al., 2013)" to "as described by Paulino et al. (2013)".

23. P. 9, equation 1. Please define precisely the meaning of Vaddition and Vmesocosm.

24. P. 11, lines 6-7. The sentence "Before presenting..." is redundant and could be removed – this is self-evident to the reader.

25. P. 11, lines 10-11. Where the correlations statistically significant?

26. P. 12. Please define the meaning of the error bars on Figure 3.

27. P. 13, line 1. Does "highest resolution" refer to temporal resolution? Please clarify.

28. P. 13, lines 14-16. Is linear regression meaningful for these data? Why use linear regression in this case?

29. P. 14, lines 2-5. What do the +/- symbols represent here?

30. P. 14, line 4. Change "measurements was" to "measurements were".

31. P. 14, line 21. There is no section 3.3.

32. P. 15. Figure 5 is unreadable as it is too small.

33. P. 15, line 11. Should this refer to Fig. 5(c) rather than Fig. 5(b)?

---

## Author Comment (AC1) · 21 Jan 2019

Two reviewers are thanked for detailed comments on the BGD text. Please find responses below (R:).

Meso and micro-cosms were sampled at different locations. The explanation of these experiments is very hard to follow. I am not sure all information is necessary, at least in the main manuscript. A simple table like table 1B is in my view enough.

R: This is obviously a matter on which there are different opinions between different co-authors and different reviewers with some wanting more detail and some wanting less. As a companion text describes the same set of mesocosms (which will be linked on the BGS website), we have reduced the method section further again but are reluctant to remove much more material.

[Figure]

From all these experiments FeII was measured and some samples were taken to observe FeII oxidation rates in the dark. In the end the explanation of different results hardly needs the differences between the experiments, the more reason to move text on the "cosms" to the supplementary information.

R: Yes the 'chemists' working on this project would certainly agree with this opinion, but others would argue that the exact setup with respect to light availability, filtering of the water, adding of zooplankton etc are all important details which may have unintended effects on Fe biogeochemistry and thus feel that it is important to include them.

The authors missed one important publication Rijkenberg et al., 2006. In this paper the influence of organic ligands of FeIII and FeII on photo-reduction and oxidation of Fe are studied. In the present text it is assumed to be important, I agree, and the Rijkenberg publication can certainly help here instead of referring to papers where speculation on this subject can be found.

R: We had missed the relevance of the Fe(II)-PPIX work in this paper because the manuscript primary concerns Fe(II) formation rates from Fe(III)-L species, but the comment on Fe(II) production in the dark at the end of the text referred to are indeed very useful and compliment the comments from Reviewer 2 concerning the possibility of 'dark' Fe(II) production from a superoxide driven redox cycle. An additional paragraph is added in the discussion under the topic of 'dark' Fe(II) production: "Apart from the influence of organic Fe(II) ligands on Fe(II) stability arising from the slower oxidation rates of some complexed Fe(II) species, Fe(II) binding organics may also have a role in the generation of superoxide which is speculated to be a dominant mechanism for the formation of Fe(II) in the dark. Experiments with 65-130 nM of protoporphyrin IX demonstrate increased formation of Fe(II) in the dark with both increasing porphyrin concentration and increasing irradiation of seawater prior to the onset of darkness (Rijkenberg et al., 2006). Whilst the rates of this process are challenging to investigate at the sub-nanomolar porphyrin concentrations expected in natural seawater, the dark formation of Fe(II) mediated by ROS interactions with Fe(II)-organic complexes could

potentially be important in both the diurnal cycling of Fe in the surface ocean and the non-photochemical formation of Fe(II) in the dark of the ocean's interior (Rose 2012). From a mechanistic perspective, it is difficult to establish from the experiments here in whether apparent Fe(II) stability arises from reduced oxidation rates due to Fe(II) complexation, or dark Fe(II) formation via a mechanism, such as that proposed for superoxide, which involves Fe(II)-organic complexes."

One of the reasons that the text is hard to follow is the use of different names for the same locations/experiments (there are 5 locations: the Arctic (Svalbard), the Mediterranean (probably Crete), Patagonia, Gran Canaria and Kiel (mentioned only once and not in methods, but important for the discussion)). Since the setup is so complicated it is even advisable to mention experiments in the same sequence (helping the reader) and not as in the example given changing the sequence in one sentence: "MesoArc than for MesoPat (MesoPat R2 0.0022, gradient 0.0049 $\pm$ 0.014; MesoArc R." etc The introduction tells the reader that she/he can expect mesocosm experiments in three places. Then section 2.1 starts and experiments were done in 4 places en in the result section experiments in Kiel appear to have happened too. Extra names like the ocean Certain project come out of the blue. First meso is used later without any explanation on page 14 Meso Med, MicroPat MultiPat are used. These last set meso micro and multi are much better suited because they indeed indicate which kind of experiments are meant. Why not use them immediately and define them properly?

R: In hindsight this in confusing to the reader. The reason was we had used the exact terminology adopted by the projects that ran the mesocosms, but this can be confusing as 'MesoPat' was used [within the project that funded it] to refer to the field campaign (which included a mesocosm/multistressor/microcosm in Patagonia). We have therefore adopted a standardized name for each specific experiment e.g. 'MesoPat' refers to the mesocosm in Patagonia, 'MicroPat' refers to the microcosm in Patagonia etc... and adopted these throughout the text.

Section 3.5 is very interesting but difficult to read and understand. Half of the text

should be in methods, also be more clear about the Kiel experiments (at least I suspect the spiked Atlantic tests are done in Kiel).

R: As suggested (here and later), we have moved any descriptive material including equations to the method section. Yes the 'Kiel' experiments are those with Atlantic seawater, we now refer only to 'spiked Atlantic seawater experiments' And it is not clear whether the different treatments had influence on the results.

***R: This is a key point of the paper which we try to clarify better. We can't address this question (whether the different treatments had influence on the results) because the 'stability' of Fe(II) measured is very sensitive to the Fe(II) concentration at the time the experiment starts (i.e. the time at which seawater is moved into the dark). This is the fundamental finding of the paper (and is raised in some other points below, so I will address it extensively here). Conceptually, in simple terms, there are two Fe(II) 'pools' in seawater when it sits under ambient surface conditions – whether in a mesocosm, or not. A) There is a pool of inorganic Fe(II) which has oxidation rates exactly as predicted by experiments where Fe(II) is spiked into synthetic seawater. B) There is a pool of organic Fe(II) associated with natural organic material akin to ligands which, overall, has a slower oxidation rate than the inorganic pool. Therefore, when a seawater sample is sub-sampled for analysis and moved to the dark (in a bottle, or in opaque tubing flowing into a flow injection analyzer), Pool A decays faster than Pool B. The fraction of the total Fe(II) present as B therefore increases in the seconds-minutes after sample collection. Hence a major experimental problem; even if seawater is pumped straight into a flow injection analyzer using the best available method (a duel-loop FIA system), it experiences 30-60 seconds in the dark prior to analysis. In reality this time is 2-4 minutes due to the time required for subsampling, moving FIA lines, achieving stability with the luminol signal etc. . . During this time the fraction of Fe(II) present as Pool B increases. And because the half-life of pool A is short, the fractional importance of B can increase significantly within minutes of being in the dark. Furthermore, the fractional importance of A and B likely changes on diurnal timescales and over the

experiment duration. Therefore, there is a strong bias towards measuring oxidation rates close to the calculated inorganic rates at times of day when there is a large total Fe(II) pool, and when the decay rate is measured from the exact time at which a sample is moved into the dark. Conversely, there is a strong bias towards measuring apparent Fe(II) stability when the decay rate is measured at a time of day when the total Fe(II) pool is smaller, and when the decay rate is measured from some time after a sample has been moved into the dark. Unfortunately, with any sort of field experiment it's very difficult to measure more than 1 or 2 decay rate experiments simultaneously, so the time of day when an experiment was conducted varies within each experiment set, and it's impossible to produce an experimental setup free from artefacts where the sample time/'move to dark time' is completely identical to the time at which the first sample peak is measured. There is inevitably a delay and unfortunately the delay, even if varying only from 30-60 seconds, can be equivalent to 1-2 Fe(II) half-lives for inorganic Fe(II) species. For these reasons it is not meaningful to look at the difference between treatments within or between mesocosm experiments for the Fe(II) decay rate because the decay rate is biased by the initial Fe(II) concentration for the decay rate experiment and this cannot easily be corrected for.

In the discussion I miss apart from the Rijkenberg paper, discussing the influence of the different sampling and measurement treatments and the different experimental conditions.

Is the difference in time between sampling and analysis discussed, Gran Canaria is different from the others. Can this have had an effect on the results? See also above section 3.5. What is e-microcosm (in Suppl table),this is not explained, still here the largest differences between kmeas and kcal exist.

R: A line is explicitly added to discuss the potential effect of sample acidification. We provide a reference (as in the original text) which describes this in detail. We did not conduct kinetic experiments in Gran Canaria, therefore there is no direct potential for this method -change to affect our main conclusions... "a modification outlined by

Hansard and Landing (2009) which is not thought to significantly affect in-situ Fe(II) concentrations during the short time period between collection and analysis". We also note that in warm seawater there is simply no alternative as the half-life of Fe(II) limits the ability to do any analysis on unamended seawater. 'E-microcosm' was the official project name for the 'MicroPat' experiment (hence comment above, the official project 'names' are difficult to follow, so we have standardized and amended throughout). Yes this experiment shows the highest $\Delta k$. But, as noted above (***R) it is very challenging to claim this due a specific biogeochemical phenomena. Detailed comments Sections 2.1 and 2.2 are hard to follow Sentences like "Note that previously a series of experiments in the Mediterranean ('MesoMed') was also included." do not help. If it was previously, why do we bother here.

R: As raised by another reviewer, it is important to note that we attempted these measurements but do not present data (because every single measurement attempted was below detection) to avoid miss-reporting of our findings.

Line 7-9 page3 section 2.1 seem out of place, this has nothing to do with setup and sampling.

R: Moved to results section

Line 13: 10 identical 1000-1500L tanks, 5 tanks got zooplankton. According to table 1A they all received copopods but the addition was different per location.

R: No, table 1A simply lists the variables which were manipulated in each experiment. Table 1B gives the specific treatment for the high zooplankton tanks. We clarify in table B that the treatment is the zooplankton added to the 'high grazing tanks' and not the baseline for all tanks (which was zero, the tanks were filtered through a mesh, and then zooplankton were re-added to 'high' tanks only).

I did not find figure S1, below text and pictures, there is a caption but no schematic figure of the experimental design.

R: There is a table for each experiment matrix. Figure S1 is re-labelled 'Supplementary material' rather than a 'Figure'

Line 26: can bags stand? Are bags mesocosms? In the next line the word tank is used, is this still the same thing?

R: One of the mesocosm experiments used bags (Gran Canaria), two of the experiments used tanks (MesoPat/MesoArc). 'mesocosm' is used widely within the field to refer both to mesocosm experiments (i.e. the whole experiment), but also to each unit within a mesocosm experiment, hence why we try not to use both meanings in the same sentence. We have rephrased throughout now using only the term 'mesocosm experiment' to refer to a whole experiment.

Section 2.2 What is a 10-treatment?

R: an experiment with 10-treatments.

Section 2.3,line 26 after cleaning, what happened with the bottles? Were they stored empty or filled, if so with what.

R: They were stored 'empty'. Now stated explicitly.

Page 7: were the FeII bottles dark plastic?

R: No, to move Fe(II) samples from the experiments to the FIA (which took 1-2 minutes), we opted for transparent containers so the water would remain exposed to ambient light. It was then transferred into a dark box as stated in the text at a time recorded as 'time zero'. We rephrase and clarify this sentence. In Gran Canaria, where the Fe(II) samples were acidified, the bottles were opaque to prevent any further photoproduction of Fe(II) (also clarified in methods).

Line 9: Ocean certain is? It would be so much easier when the normally used names are used here too, meso/micro-Med-Arc-Pat.

R: We now use the terms Meso/micro/multi-Med/Arc/Pat throughout.

Section 2.5: What happened in Kiel? Line 18 tells us what happened in Patagonia and Svalbard (? Pat and ARC, Comau fjord- Kongsfjorden?)

R: we now explicitly add the names (meso/micro-Med-Arc-Pat) when referring to any fieldsite and refer to the laboratory spiked experiments consistently as spiked seawater experiments.

In 3.5 79 experiments are mentioned, that info belongs here. How many per location. Were they kept under ambient temperature, where is the laminar flowhood

R: Details added to the methods section. All of these experiments were in temperature controlled rooms with temperatures as per the collected water for analysis. Page 8 s2.5 re-written accordingly, . . . 'All Fe(II) decay experiments were conducted inside the temperate controlled rooms hosting the MultiPat/MultiArc experiments. As such, a constant temperature was maintained throughout these experiments. The FIA instrumentation was arranged with the inflow lines under a laminar flow hood inside the temperature controlled rooms. . ..'

3.1 why use the name Svalbard here and not MesoArc.

R: Ammended.

Page 10 line 11, no glucose is mentioned in Table 1

R: No, this was only in the text, now amended.

Page 11 line 9: curiously..Why.is this curious, and why give relations that are not relations, for figures 2 and 4. For figure 4 it is not clear which line belongs to which mesocosm experiment. No idea what the journal guide lines are, but especially as in figure 3the './ 'is confusing like there is a ratio instead of the unit.

R: Because we can't think of a simple reason why there should be a strong correlation between DFe and TdFe in one mesocosm, but no correlation at all in another, especially when the strongest correlation is for a low organic carbon, high particulate Fe

site. Figures 2 and 4 are removed to save space. Units are changes on the graphs throughout.

Page 13 line 1 as per?

R: There are obviously multiple manuscripts in preparation from this series of meso/multi/micro experiments. One concerns H2O2 (in BGS, the manuscripts will be linked as companion papers so the link will be more obvious).

Page 14: Meso Med, MicroPat MultiPat: have mercy on your reader! Suddenly new abbreviations. However, useful abbreviations that should be used throughout the whole manuscript

R: We have now adopted our own standardized terms for the experiments as per previous comments.

Lines 9-10: Is that to be expected? Reference needed here (also at line 16)

R: Yes, this is discussed in the next paragraph (in the original text). A Reference is added for Fe:P lithogenic material and high TdFe at this fieldsite.

9-16, a lot of different names for the same sites

R: Yes, but each mesocosm obviously has a corresponding fieldsite so we inevitably have to name the experiment and the site. For clarity we annotate the place names with the experiments hosted at that location.

29: chlorophyll a Italic Figures 5 are too small. Lables and legend are impossible to read and the sequence in the legend is not logic, one legend might be enough for 5a and 5b. Perhaps make 5 c a separate figure. Be careful with ratio's. The high values are they due to low DFe? Figure 5 c is not mentioned in the text.

R: Amended so the figures display better whilst merged in the text. The legend sequence order is changed. C is shown separately and we now show the concentrations and the ratio (DFe is never particularly low, so no these are not an artefact of subnanomolar DFe concentrations).

Line26: I do not know whether there is a general decline as claimed here, the 1450 microatm perhaps does decreases but the 1300, 700 and 1150 microatm do not, so no general decline here. This figure is not suited to make such statements. An increase between days 20-29 ok.

R: The significance of changes (other than the increase after day 20) is variable between treatments, as this doesn't really affect our conclusions we slim the text accordingly and remove these lines.

Page 15: line 6: what is number 7?

R: Changed to PCO2 value for this mesocosm.

Lines 6-9: not clear what the authors tell here? Is this still about figure 5b? The sixth mentioned number does not show an increase, not the seventh. They have different CO2, haven't they?

R: We clarified this 'number' (above) referred to a specific treatment. It is the 6th number because one mesocosm leaked and was removed from the experiment (this is all clarified by just referring to the treatments by PCO2 target level throughout rather than arbitrary treatment labels).

Page 16, Line 11: which variation, give reference, do not force the reader to search in another of your papers to find out.

R: The specific variations compared were a dual-loop configuration as described by Croot and Laan (2002) and a pre-concentration method as described by Bowie et al., (2002)

Lines12-14: it depends what you mean with in situ and what you want to do with the k-values. With such an uncertainty one can wonder whether waiting for stabilisation would have been wiser.

[Figure]

R: But then any Fe(II) present would have decayed and we would have to conducted spiked experiments, which wouldn't tell us much about stabilization that we didn't already know. This error is not large considering comparable data in the literature considering what reported 'errors' actually include. Note the exact opposite query is raised by reviewers with Sarthou et al., (2011) BGS. In this excellent 2011 manuscript the authors added small Fe(II) spikes to seawater in order to determine oxidation rates and it was questioned (see comments/discussion with reviewer 2 on that text) whether this approach was meaningful compared to observing in-situ decay rates. Hence our rationale for the setup herein.

Lines 20 onwards: is this what happened in Kiel? Most of this belongs to the method section. Also the equations belong to the method section in my view.

R: Yes, we have moved equations and experiment descriptions to the methods section.

Page 17:I can add a few hypotheses: the aged Atlantic water was probably filtered, in any case no phytoplankton or copepods were present. The added Fe for certain saturated the organic ligands and thus this DFe was in an inorganic form, a colloidal or amorphous Fe-oxide or hydroxide. This is where the equations 2 and 3 were made for: inorganic Fe! So certainly this is other chemistry. I advise to read Rijkenberg et al., 2006 Geochimica et Cosmochimica Acta 70 (2006) 2790–2805; Enhancement and inhibition of iron photoreduction by individual ligands in open ocean seawater.

R: Yes, these experiments represent inorganic speciation and this is exactly what we summarize in hypothesis III. Note the experiments concerning ligand saturation in the Rijkenberg 2006 text concern Fe(II) formation from Fe(III) species, a slightly different issue to the Fe(II) decay discussed here.

Page 18:line one, why would low FeII be the most stable?

R: See *** comment above; because the fraction of this Fe(II) existing in a stable form is likely higher.

Discussion line 18: This can be read that TFe behaves conservatively. Why would DFe-TFe be linear, that is a strange idea. That is assuming all particles have the same properties.

R: In these near-shore waters where a large fraction of Fe comes from a near-point source (e.g. the freshwater outflow into the Pat/Arc fjord sites) it is not an unreasonable statement that all particles have the same properties, note the relative consistency in XRF data, simply because freshwater derived particles account for the vast majority of Fe-rich particles in the water column. On short timescales TdFe does behave conservatively, unlike the rapid removal of DFe in these nearshore environments, a TdFe/S plot is linear showing that the sinking/modification of TdFe takes longer than the residence time of water in these fieldsites.

Page 19: table 4 why is mesopat meso and multistressor so different, this is not discussed. Why is the sequence different, why Svalbard whereas it is Arctic. The different names makes it more difficult to understand.

R: Noting the large standard deviation on both Fe(II)/DFe and DFe/TdFe, it is not clear that they are 'so different'. We clarify that the 'ambient' measurements are at the fieldsite. But as the 'ambient' measurements don't refer to any of the experiments at that fieldsite, they need a separate name. For clarity we sate 'Arctic (Svalbard)'.

Lines 23-24: why was this not mentioned in the method section?

R: The exact timing of the experiments is shown in the data table appended to the paper and thus we don't feel it necessary to write it out in the methods section. The aim of these experiments was to investigate how Fe(II) decayed, not to produce high resolution Fe(II) time-series across the duration of every mesocosm.

Page 20 line 8-9: thus what is the conclusion?

R: A line is added, 'The ambient concentrations of Fe(II) measured in Patagonia (Comau fjord) and the Arctic (Svalbard, Kongsfjorden) at the mesocosm experiment field-

sites are therefore not necessarily directly comparable to Fe(II) concentrations measured after nutrient addition in the mesocosm experiments.' 4.3: line 16: according to the methods section artificial light was used in micro and multistressor but not in Mesocosm, so why mention artificial light here?

R: The text states 'due to the enclosed HDPE mesocosm design and/or synthetic lighting'. The point being made was that all of the mesocosm/microcosm/multistressor experiments where Fe(II) decay experiments were conducted had low $H_2O_2$ concentrations.

Lines 21-25: read Rijkenberg, they saw the influence of ligands on Fe redox, of ligands binding Fe III en of a ligand binding FeII. That should be added in the discussion here.

R: Photochemical formation of Fe(II) from Fe(III)-L species is not relevant to the discussion here. The specific points concerning dark Fe(II) formation from porphyrin are however interesting and added at the end of this section as per some earlier comments.

Page 21decay rates in the e-microcosm are different from the calculated k compared to the others, apart from low FeII at t=0? (low Fe(II) occurs also in other experiments) what is e-microcosm, what is different? Could that be an extra reason. Use the work of Rijkenberg et al in the discussion on page 21, they did not assume FeII ligands, they used one in their redox rate experiments.

R: (E-microcosm is the data label for the MicroPat experiment, we have changed this in the text to 'MicroPat' as per our standardized names). There is no obvious experimental difference between the MesoPat/MicroPat/MultiPat experiments that immediately provides an easy explanation for why the largest changes in K should be reported for datapoints from one experiment. There may of course be species-level effects due to the different biological communities at the start of, and throughout, each experiment. Yet, as we note (R***), because the discrepancy between measured and calculated K is very sensitive to the Fe(II) concentration at t=0, and because it is not (using the design here) possible to rigorously standardize t=0 so that [Fe(II)] at t=0 is constant, or

to account for the change in Fe(II) concentration and speciation between in-situ conditions and t=0, it is very difficult to deduce any relationships between biogeochemical parameters and the difference in K.

Excel file temp in k, make capital, add start or initial also to the column name for FeII. The precision does not warrant the decimals shown with 35% uncertainty. What is an e-microcosm, why are the rates so high here. Add measured to k. No Kiel experiments here?

R: Amended. (See above comment also). 'Kiel' (spiked Fe(II) decay experiments) data is now added to the supplementary file as per the Meso/Micro/Multi data.

---

## Author Comment (AC2) · 21 Jan 2019

Two reviewers are thanked for detailed comments on the BGD text. Please find responses below (R:).

This manuscript addresses a topic that is relevant to the scope of Biogeosciences. There is a clear rationale for the work, the experiments have been carefully designed and executed, and the data analysis and interpretation are reasonable. However, I felt that the scope of the paper needs to be more accurately represented and that some details relating to the experiments and data analysis were missing. Overall, I believe that this manuscript should be published after revision. SPECIFIC COMMENTS 1. The title of the manuscript is too broad, to the extent that I fiĄnd it misleading. The manuscript does not directly address the issue of Fe(II) stability in seawater – there are no measurements of thermodynamic constants (which the word "stability" implies),

nor measurements or calculations of complex speciation, the underlying mechanisms are inferred or hypothesised rather than explicitly measured or tested, and the measurements are limited primarily to coastal seawater. This is all perfectly valid, but the manuscript really addresses iron redox speciation in coastal mesocosm experiments, and I would prefer to see a title more along these lines.

R: New title suggested: "Fe(II) stability in coastal seawater during experiments in Patagonia, Svalbard and Gran Canaria." We accept that, to a chemist, 'stability' would imply thermodynamic stability, but in an environmental context -and considering how the term is used in prior Fe(II) work- it is challenging to find an alternative phrase within a limited title word count.

2. The assertions about "over-use of the "99%" statistic" (i.e. that "99% of DFe in the oceans is hypothesized to be present as Fe(III)-complexes" are subjective and I find this aspect of the Introduction to be overstated. It is true that "this observation explicitly or implicitly underpins the formulation of DFe in global marine biogeochemical models", and that the influence of Fe redox speciation is often ignored. The authors also provide a nice summary of compelling evidence that Fe(II) is important in "two specific environments". However, it does not automatically follow that the assumption that 99% of DFe is present as Fe(III)-complexes is invalid everywhere in the oceans, or that the "99% statistic" is "over-used". To make this assertion objectively would require something like a meta-analysis of the literature to quantify the number of papers that make this claim, and the proportion of those that make this claim incorrectly. In my opinion, it would be better to just present the evidence and let the reader decide if they think this is an "over-used" statistic. I would suggest that the authors review the Introduction to remove or tone down subjective statements and ensure that any assertions are supported by an appropriate number of references.

R: Agreed, as the text is already quite long we have no desire to extent it further with an unnecessary literature review. Given that the key point is that all global biogeochemical models represent dissolved Fe basically as a dissolved Fe(III)-L species without

the complication of a redox cycle, we delete the line in question, ['Yet, as evidenced by over-use of the "99%" statistic, the presence of a fraction of DFe as Fe(II) in surface waters –exactly where most primary production occurs- is widely overlooked.]' And, as per other comments, this is replaced with a brief over-view of Fe(II) measurements in the deep ocean. 'Fe(II) concentrations at depth are less well characterized, although there is extensive evidence of pM Fe(II) concentrations occurring throughout the pelagic water column suggesting that 'dark' Fe(II) production is a widespread phenomenon (Sarthou et al., 2011, Sedwick et al., 2014, Schallenberg et al., 2015).'

3. In the Introduction there is a strong focus on why Fe(II) is important, but the background about what is known in relation to the abundance and behaviour of Fe(II) in the ocean seems incomplete. For example, the growing body of work (including by some of the co-authors of this manuscript) around the influence of organic exudates from marine phytoplankton on Fe(II) oxidation kinetics is not mentioned in the Introduction, but this would seem critical to understanding much of the manuscript and its rationale. In addition, while there is a brief overview of Fe(II) dynamics in the photic zone and in suboxic zones, it would also be useful to briefly review reports of Fe(II) measurements in other parts of the ocean.

R: An extensive discussion of the role of organics on Fe(II) 'stability' is included in the discussion (as per the original text). We now also include a few lines of introduction to this subject in the introduction. 'Fe(II) speciation in seawater and the potential role of ligands in Fe(II) biogeochemistry is however still uncertain. Organic Fe(II) ligands, akin to Fe(III) ligands in seawater but likely with different binding constant ranges and functional groups (Boukhalfa and Crumbliss 2002), are widely speculated to affect the oxidation rate of Fe(II) in seawater (Santana-Casiano et al., 2000, Rose and Waite 2003, Gonzalez et al 2014). Yet characterizing the concentration and properties of organic Fe(II) ligands in natural waters using titration approaches, as successfully adapted to determine Fe(III)-speciation, has proven challenging (Statham et al., 2012) due to practical difficulties in stabilizing Fe(II) concentrations without unduly affecting its

speciation. Never-the-less a broad range of cellular exudates have been demonstrated to positively affect Fe(II) concentrations in seawater, either via enhancing Fe(II) formation rates or retarding its oxidation rate (Rijkenberg et al., 2006., Santana-Casiano et al., 2014, Lee et al., 2017).'

4. Analysis of Fe(II) data was based on an assumption of pseudo-first order kinetics, but there are no details on whether this assumption was tested or verified.

R: This is indeed assumed here, as elsewhere in manuscripts on the same topic, but also demonstrated to be a reasonable assumption with the linearity of the ln[Fe(II)] vs time response for each experiment where data is presented (these values are already included in the datasheet). This is clarified in the main text . . .. 'correlation coefficients are noted for each linear regression'. . ..

5. I think it is highly problematic to exclude discussion of the Mesomed Fe(II) results from the manuscript because "Fe(II) concentrations were universally < 0.2 nM" (p. 3, lines 8-9). Given that you are arguing that Fe(II) is widespread and overlooked, excluding presentation of results from one set of mesocosms because Fe(II) was not measurable in those conditions could be perceived as cherry picking data. Again, I think this would be less of an issue if the scope of the manuscript as suggested by the title and Introduction was revised. If this is recast to make it clear that this is a study of Fe(II) dynamics in a discrete set of mesocosm experiments, then I think it is fine to mention the Mesomed experiments in this way without a detailed presentation of results. However, I think it is also important not to overlook these results in the discussion when generalising about Fe(II) behaviour.

R: 0.2 nM was the detection limit. So it isn't the case that we excluded results, not a single [Fe(II)] for any of the Med experiments was above the detection limit of 0.2 nM. This may simply reflect the high temperature of the Med experiments (20°C) and similarly unfavorable pH/Salinity for Fe(II) measurements; the half-life of Fe(II) under in-situ Med conditions was sufficiently short that it would be practically impossible to

measure in situ Fe(II) concentrations with a dual-loop FIA system as multiple Fe(II) half-lives occur as sample water is flowing into the FIA. There isn't therefore any insight to be gained from the Med work. The text is changed slightly to address this 'Fe(II) concentrations were universally below detection <0.2 nM. . ..'

6. The discussion about processes contributing to Fe(II) formation lacks mention of superoxide-mediated Fe reduction or other biological ferrireductase processes. This would seem remiss given that recent publications have suggested extracellular super-oxide production may well be ubiquitous(e.g. Diazetal., 2013, Widespread production of extracellular superoxide by heterotrophic bacteria, Science 340: 1223-1226) and is likely to inïnĆuence Fe speciation (e.g. Rose, 2012, The inïnĆuence of extracellu-lar superoxide on iron redox chemistry and bioavailability to aquatic microorganisms, Frontiers in Microbiology 3:124).

R: As noted, we do not we do specifically investigate the mechanism of apparent Fe(II) stability, but the potential role of O2- is certainly of interest in light of the Rijkenberg 2006 work highlighted by another reviewer. A paragraph addressing this point of in-terest is added at the end of the discussion. A ubiquitous 'background' production of radicals in the deep ocean by bacteria would indeed be interesting as a potential driver of trace element redox chemistry, but we note it is incredibly challenging to make re-liable measurements of trace species under dark pelagic (i.e. below the photic zone) conditions and thus very speculative to comment on the potential significance of O2-/Fe cycling on a grand scale; 'Apart from the influence of organic Fe(II) ligands on Fe(II) stability arising from the slower oxidation rates of some organically complexed Fe(II) species, Fe(II) binding organics may also have a role in the generation of superoxide (O2-) which is speculated to be a dominant mechanism for the formation of Fe(II) in the dark. Experiments with 65-130 nM of protoporphyrin IX demonstrated increased formation of Fe(II) in the dark with both increasing porphyrin concentration and in-creasing irradiation of seawater prior to the onset of darkness (Rijkenberg et al., 2006). Whilst the rates of this process are challenging to investigate at the sub-nanomolar

porphyrin and Fe(II) concentrations expected throughout most of the ocean, the dark formation of Fe(II) mediated by ROS interactions with Fe(II)-organic complexes could potentially be important in both the diurnal cycling of Fe in the surface ocean and the non-photochemical formation of Fe(II) in the dark of the ocean's interior (Rose 2012). From a mechanistic perspective, it is difficult to establish from the experiments herein whether apparent Fe(II) stability arises from reduced oxidation rates due to Fe(II) complexation, or dark Fe(II) formation via a mechanism, such as that proposed for superoxide, which involves Fe(II)-organic complexes.'

7. The organisation of different aspects of the manuscript needs to be reviewed to ensure material is presented in the correct location. For example, the first paragraph of section 3.1 is discussion, not results. The second paragraph of section 3.1 is methods, not results.

R: Text shifted accordingly. A greater number of subtitles are now used to separate the method/results/discussion of each component.

Details about measurement of hydrogen peroxide concentrations are n provided in the methods section at all, but rather addressed only by the statement "as per Hopwood, 2018" in the results section.

R: A text mainly concerning H2O2 in the mesocosms is also under review for BGS, the two are linked as companion manuscripts and thus we did not want to include unnecessary detail in this already long manuscript.

8. P. 1, line 14. I suggest changing "exclusively" to "almost exclusively" or "primarily". It is not strictly correct to say that dissolved Fe speciation is assumed to consist exclusively of Fe(III)-L, as Fe' is generally also considered (although minor).

R: 'Almost exclusively' used where applicable.

9. P. 2, lines 31-32. The argument that "the potentially widespread presence of Fe(II)" implies that "redox cycling is an important feature of marine Fe biogeochemistry" is

a circular argument. The three cited papers do not show that Fe(II) is potentially widespread – they show that Fe(II) is persistent in certain specifić environments and locations studied in this papers. I don't mean to be overly critical about this – I think Fe(II) is important and possibly overlooked – but I think it's important to be objective and precise.

R: Rephrased . . ..'raise interest in the role of redox cycling in the marine biogeochemical Fe cycle.'

10. P. 17, lines 9-11 and 19-21. This hypothesis is not plausible, in my opinion. A difference in rate constants between different Fe(II) concentrations could be related to a difference in chemical mechanism, but should be completely independent of calibration. Also, there are several studies of Fe(II) oxidation kinetics in seawater that have been conducted at low nanomolar concentrations such that there is a coherent mechanistic understanding (and ability to predict) Fe(II) rate constants from the low nanomolar range right through to the micromolar range.

R: Yes there are multiple studies providing excellent formulas for the calculation of the rate constant with varying T/S/pH/O2. Constructing spreadsheets from different formulations does produce small changes in the calculated value of K (or t1/2) which are systematic. These are however minor. Here we opted to use a single, already published, formulation to determine K and we agree that under these conditions (T/pH/O2)-which are generally well covered by experimental rate constant data,- there is low uncertainty in the value of K. But we thought it was important to raise, and eliminate the suggestion nevertheless.

TECHNICAL CORRECTIONS 11. P. 1, line 20. I suggest changing "retarded relative to" to "less than". Rates can be fast or slow, but rate constants are large or small.

R: changed.

12. P. 1, line 25. Please add a qualiffcation to this sentence explaining under what

conditions your work challenges these assumptions (e.g. in coastal surface waters?).

R: 'in coastal seawater' added.

13. P. 2, line 8. Ligands are not necessarily small or organic. Perhaps could change this to "Organic ligands (L) capable of complexing Fe(III) can..."

R: 'organic' is added to the prior sentence to clarify. We define ligands as 'small' and 'organic' when referring to filtered DFe in seawater . The supporting references demonstrate that these ligands are organic.

14. P. 2, lines 24-26. 14. This sentence seems like it belongs in the next paragraph... I can't see how this relates to the presence of Fe(II) in suboxic or photic zones.

R: lines now separated.

15. P. 2, line 28. "There is a paucity of Fe(II) data..." – what sort of Fe(II) data?

R: amended 'pelagic Fe(II) concentration'

16. P. 2, line 29. What do you mean by "kinetic availability"? Do you mean kinetic lability?

R: Yes, but specifically in the context of cellular uptake. The kinetic lability of Fe(II) makes its uptake (theoretically) less energetically costly than Fe(III) uptake. (We now use 'lability' to avoid ambiguity). But this is an over-simplistic argument because cellular-uptake systems may be specifically designed to bind Fe(III) at the cell surface, an argument we don't wish to raise here, hence we simply use the Sunda reference to state that it is theoretically more favorable for a cell to uptake Fe(II) than Fe(III) from a simple energetic perspective.

17. P. 2, lines 34-35. "as evidenced by over-use of the 99% statistic" – what evidence? No citations are provided and this is not tested robustly, as stated in point 2 above.

R: Rephrased to refer exclusively to the use for a formulation based on this assumption

in global biogeochemical models (as above).

18. P. 3, lines 6-10. Following on from point 5 above, I find it confusing that some Mesomedresultsareincludedintheresults, butnodetailsareprovidedinthemethods about these experiments, other than these couple of sentences. I think you need to treat this dataset in a similar way to the other mesososm results, namely describe the method details in full, and fully account for the Mesomed results in your discussion.

R: Rephrased for clarity, there are no results, all data was below detection due to the challenge of measuring Fe(II) using the setup herein under warm conditions due to the shorter half-life of Fe(II).

19. Tables 1A and 1B. It would make more sense to me to label these Table 1 and Table 2, as they show quite separate information. Furthermore, it would be useful to provide coordinates for the mesocosm locations in Table 1.

R: Done. Now provided.

20. P. 6, line 4. Can you provide any information about the spectral quality of the lighting?

R: We can provide the exact manufacturer's description which includes the wavelength distribution of the lamps.

21. P. 6, line 25. Should this be "trace metal clean low density polyethylene" rather than "trace metal low density polyethylene"? 22. P. 7, line 22. Change "as described by (Paulino et al., 2013)" to "as described by Paulino et al. (2013)".

R: Yes, amended.

23. P. 9, equation 1. Please define precisely the meaning of Vaddition and Vmeso-cosm.

R: Moved into methods and defined.

24. P. 11, lines 6-7. The sentence "Before presenting..." is redundant and could be removed – this is self-evident to the reader.

R: Removed.

25. P. 11, lines 10-11. Where the correlations statistically signifi̧cant?

R: For one experiment yes, for the other no. (Test / p-values added in text).

26. P. 12. Please defi̧ne the meaning of the error bars on Figure 3.

R: Error bars defined (always standard deviation of 3 measurements).

27. P. 13, line 1. Does "highest resolution" refer to temporal resolution? Please clarify.

R: Clarified, yes, "highest temporal resolution over the experiment duration".

28. P. 13, lines 14-16. Is linear regression meaningful for these data? Why use linear regression in this case?

R: This section didn't add much value to the text and following comments from both reviewers the figure and corresponding paragraph are removed.

29. P. 14, lines 2-5. What do the +/- symbols represent here? 3

R: Standard deviations (now explicitly stated when used and when referring to mean +/- SD for a dataset, n is specified in the text).

0. P. 14, line 4. Change "measurements was" to "measurements were". 31. P. 14, line 21. There is no section 3.3.

R: Amended.

32. P. 15. Figure 5 is unreadable as it is too small. 33. P. 15, line 11. Should this refer to Fig. 5(c) rather than Fig. 5(b)?

R: Re-structured so the figure is clearer when displayed in word.

---

## Author Response (AR2)

Dear Editor,

Following very minor comments the following changes have been made to the final revised text,

- The methods section is slimmed by removing tabulated material also included in the companion text and associated wording.
- An additional paragraph, and a few extra lines concerning the specific purpose of this manuscript, are added in the introduction (new lines 75-87). (Pasted below)

To what extent the behaviour of dissolved Fe(II) in the ocean is influenced by organic material, rather than solely by inorganic chemical processes, is however challenging to determine given the sub-nanomolar Fe(II) concentrations present in pelagic environments, and the diurnal variability expected in surface Fe(II) concentrations (Johnson et al., 1994; Miller and Kester, 1994). The clear effects of biologically derived organic material on Fe(II) stability however do indicate that aquatic micro-organisms may directly or indirectly influence the speciation and stability Fe(II) in solution around them and thus moderate the bioavailability and bioaccessibility of dissolved Fe to cellular uptake systems (Croot et al., 2001; Shaked et al., 2002; Samperio-Ramos et al., 2018a). In order to gain insight into the potential interactions between Fe(II) and marine micro-organisms in dynamic surface waters, here we adapted flow injection apparatus to measure in situ Fe(II) concentrations both in a series of mesocosm experiments (Gran Canaria, Patagonia, Svalbard) and in adjacent ambient waters. The experimental gradients in these experiments included pH, zooplankton density and dissolved organic carbon (DOC) concentration facilitating an investigation of Fe(II) stability under a diverse range of biogeochemical conditions. By allowing ambient Fe(II) concentrations to decay in the dark, the stability of in situ Fe(II) concentrations was assessed and compared to calculated inorganic oxidation rates.

- Minor corrections are made to the labels for figures 1 and3, and Tables 3 and 4 (now Tables 1 and 2)

Sincerely,

Mark Hopwood, Corresponding author

[revised manuscript text omitted]